# Meta-learning with an Adaptive Task Scheduler

**Huaxiu Yao**[1]*, **Yu Wang**[2], **Ying Wei**[3], **Peilin Zhao**[4]
**Mehrdad Mahdavi**[5], **Defu Lian**[2], **Chelsea Finn**[1]
[1]Stanford University, [2]University of Science and Technology, [3] Tencent AI Lab
[4]Pennsylvania State University, [5]City University of Hong Kong
[1]{huaxiu,cbfinn}@cs.stanford.edu, [2]{wangyu,liandefu}@ustc.edu.cn
[3]yingwei@cityu.edu.hk, [4]masonzhao@tencent.com, [5]mzm616@psu.edu

## Abstract

To benefit the learning of a new task, meta-learning has been proposed to transfer a well-generalized meta-model learned from various meta-training tasks. Existing meta-learning algorithms randomly sample meta-training tasks with a uniform probability, under the assumption that tasks are of equal importance. However, it is likely that tasks are detrimental with noise or imbalanced given a limited number of meta-training tasks. To prevent the meta-model from being corrupted by such detrimental tasks or dominated by tasks in the majority, in this paper, we propose an adaptive task scheduler (**ATS**) for the meta-training process. In ATS, for the first time, we design a neural scheduler to decide which meta-training tasks to use next by predicting the probability being sampled for each candidate task, and train the scheduler to optimize the generalization capacity of the meta-model to unseen tasks. We identify two meta-model-related factors as the input of the neural scheduler, which characterize the difficulty of a candidate task to the meta-model. Theoretically, we show that a scheduler taking the two factors into account improves the meta-training loss and also the optimization landscape. Under the setting of meta-learning with noise and limited budgets, ATS improves the performance on both miniImageNet and a real-world drug discovery benchmark by up to 13% and 18%, respectively, compared to state-of-the-art task schedulers.

## 1   Introduction

Meta-learning has emerged in recent years as a popular paradigm to benefit the learning of a new task in a sample-efficient way, by meta-training a meta-model (e.g., initializations for model parameters) from a set of historical tasks (called meta-training tasks). To learn the meta-model during meta-training, the majority of existing meta-learning methods randomly sample meta-training tasks with a uniform probability. The assumption behind such uniform sampling is that all tasks are equally important, which is often not the case. First, some tasks could be noisy. For example, in our experiments on drug discovery, all compounds (i.e., examples) in some target proteins (i.e., tasks) are even labeled with the same bio-activity value due to improper measurement. Second, the number of meta-training tasks is likely limited, so that the distribution over different clusters of tasks is uneven. There are $2,523$ target proteins in the binding family among all $4,276$ proteins for meta-training, but only $55$ of them belong to the ADME family. To fill the gap, we are motivated to equip the meta-learning framework with a task scheduler that determines which tasks should be used for meta-training in the current iteration.

Recently, a few studies have started to consider introducing a task scheduler into meta-learning, by adjusting the class selection strategy for construction of each few-shot classification task [17, 19],

---

*H. Yao and Y. Wang contribute equally; correspondence to: Y. Wei

35th Conference on Neural Information Processing Systems (NeurIPS 2021).

directly using a self-paced regularizer [3], or ranking the candidate tasks based on the amount of information associated with them [11]. While the early success of these methods is a testament to the benefits of introducing a task scheduler, developing a task scheduler that adapts to the progress of the meta-model remains an open challenge. Such a task scheduler could both take into account the complicated learning dynamics of a meta-learning algorithm better than existing manually defined schedulers, and explicitly optimize the generalization capacity to avoid meta-overfitting [23, 30].

To address these limitations, in this paper, we propose an **A**daptive **T**ask **S**cheduler (**ATS**) for meta-learning. Instead of fixing the scheduler throughout the meta-training process, we design a neural scheduler to predict the probability of each training task being sampled. Concretely, we adopt a bi-level optimization strategy to jointly optimize both the meta-model and the neural scheduler. The meta-model is optimized with the sampled meta-training tasks by the neural scheduler, while the neural scheduler is learned to improve the generalization ability of the meta-model on a set of validation tasks. The neural scheduler considers two meta-model-related factors as its input: 1) the loss of the meta-model with respect to a task, and 2) the similarity between gradients of the meta-model with respect to the support and query sets of a task, which characterize task difficulty from the perspective of the outcome and the process of learning, respectively. On this account, the neural scheduler avoids the pathology of a poorly generalized meta-model that is corrupted by a limited budget of tasks or detrimental tasks (e.g., noisy tasks).

The main contribution of this paper is an adaptive task scheduler that guides the selection of meta-training tasks for a meta-learning framework. We identify two meta-model-related factors as building blocks of the task scheduler, and theoretically reveal that the scheduler considering these two factors improves the meta-training loss as well as the optimization landscape. Under different settings (i.e., meta-learning with noisy or a limited number of tasks), we empirically demonstrate the superiority of our proposed scheduler over state-of-the-art schedulers on both an image classification benchmark (up to $13\%$ improvement) and a real-world drug discovery dataset (up to $18\%$ improvement). The proposed scheduler demonstrates great adaptability, tending to 1) sample non-noisy tasks with smaller losses if there are noisy tasks but 2) sample difficult tasks with large losses when the budget is limited.

## 2 Related Work

Meta-learning has emerged as an effective paradigm for learning with small data, by leveraging the knowledge learned from previous tasks. Among the two dominant strands of meta-learning algorithms, we prefer gradient-based [4] over metric-based [26] for their general applicability in both classification and regression problems. Much of the research up to now considers all tasks to be equally important, so that a task is randomly sampled in each iteration. Very recently, Jabri et al. [8] explored unsupervised task generation in meta reinforcement learning according to variations of a reward function, while in [11, 20] a task is sampled from existing meta-training tasks with the probability proportional to the amount of information it offers. Complementary to these methods specific to reinforcement learning, a difficulty-aware meta-loss function [15] and a greedy class-pair based task sampling strategy [17] have been proposed to attack supervised meta-learning problems. Instead of using these manually defined and fixed sampling strategies, we pursue an automatic task scheduler that learns to predict the sampling probability for each task to directly minimize the generalization error.

There has been a large body of literature that is concerned with example sampling, dating back to importance sampling [12] and AdaBoost [5]. Similar to AdaBoost where hard examples receive more attention, the strategy of hard example mining [25] accelerates and stabilizes the SGD optimization of deep neural networks. The difficulty of an example is calibrated by its loss [16], the magnitude of its gradient [31], or the uncertainty [2]. On the contrary, self-paced learning [13] presents the examples in the increasing order of their difficulty, so that deep neural networks do not memorize those noisy examples and generalize poorly [1]. The order is implemented by a soft weighting scheme, where easy examples with smaller losses have larger weights in the beginning. More self-paced learning variants [9, 28] are dedicated to designing the scheme to appropriately model the relationship between the loss and the weight of an example. Until very recently, Jiang et al. [10] and Ren et al. [24] proposed to learn the scheme automatically from a clean dataset and by maximizing the performance on a hold-out validation dataset, respectively. Nevertheless, task scheduling poses more challenges than example sampling that these methods are proposed for.

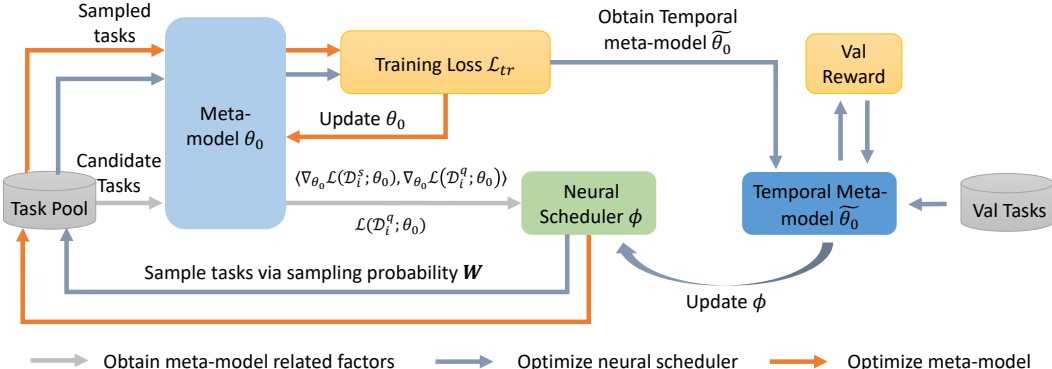

Figure 1: Illustration of ATS: (1) ATS calculates the meta-model-related factors for each candidate task (i.e., grey arrow). (2) ATS leverages the neural scheduler to sample tasks from the candidate tasks and use them to learn the temporal meta-model $\tilde{\theta}_0$, which is in turn used to optimize the neural scheduler according to the feedback from validation tasks (i.e., blue arrow). (3) The updated neural scheduler is used to resample the tasks and update the meta-model (i.e., orange arrow).

## 3   Preliminaries and Problem Definition

Assume that we have a task distribution $p(\mathcal{T})$. Each task $\mathcal{T}_i$ is associated with a data set $\mathcal{D}_i$, which is further split into a support set $\mathcal{D}_i^s$ and a query set $\mathcal{D}_i^q$. Gradient-based meta-learning [4], which we focus on in this work, learns a well-generalized parameter $\theta_0$ (a.k.a., meta-model) of a base predictive learner $f$ from $N$ meta-training tasks $\{\mathcal{T}_i\}_{i=1}^N$. The base learner initialized from $\theta_0$ adapts to the $i$-th task by taking $k$ gradient descent steps with respect to its support set (inner-loop optimization), i.e., $\theta_i = \theta_0 - \alpha \nabla_\theta \mathcal{L}(\mathcal{D}_i^s; \theta)$. To evaluate and improve the initialization, we measure the performance of $\theta_i$ on the query set $\mathcal{D}_i^q$ and use the corresponding loss to optimize the initialization as (out-loop optimization):

$$\theta_0^{(k+1)} = \theta_0^{(k)} - \beta \sum_{i=1}^N \mathcal{L}(\mathcal{D}_i^q; \theta_i), \tag{1}$$

where $\alpha$ and $\beta$ denote the learning rates for task adaptation and initialization update, respectively. After training $K$ time steps, we learn the well-generalized model parameter initializations $\theta_0^*$. During the meta-testing time, $\theta_0^*$ can be adapted to each new task $\mathcal{T}_t$ by performing a few gradient steps on the corresponding support set, i.e., $\theta_t = \theta_0^* - \alpha \nabla_\theta \mathcal{L}(\mathcal{D}_t^s; \theta)$.

In practice, without loading all $N$ tasks into memory, a batch of $B$ tasks $\{\mathcal{T}_i^{(k)}\}_{i=1}^B$ are sampled for training at the $k$-th meta-training iteration. Most of existing meta-learning algorithms use the uniform sampling strategy, except for a few recent attempts towards manually defined task schedulers (e.g., [11, 17]). Either the uniform sampling or manually defined task schedulers may be sub-optimal and at a high risk of overfitting.

## 4   Adaptive Task Scheduler

To address these limitations, we aim to design an adaptive task scheduler (ATS) in meta-learning to decide which tasks to use next. Specifically, as illustrated in Figure 1, we design a neural scheduler to predict the probability of each candidate task being sampled according to the real-time feedback of meta-model-based factors at each meta-training iteration. Based on the probabilities, we sample $B$ tasks to optimize the meta-model and the neural scheduler in an alternating way. In the following subsections, we detail our task scheduling strategy and bi-level optimization process.

### 4.1   Adaptive Task Scheduling Strategy

The goal for this subsection is to discuss how to select the most informative tasks via ATS. We define the scheduler as $g$ with parameter $\phi$ and formulate the sampling probability $w_i^{(k)}$ of each candidate task $\mathcal{T}_i$ at training iteration $k$ as

$$w_i^{(k)} = g(\mathcal{T}_i, \theta_0^{(k)}; \phi^{(k)}), \tag{2}$$

where $w_i^{(k)}$ is conditioned on task $\mathcal{T}_i$ and the meta-model $\theta_0^{(k)}$ at the current meta-training iteration. To quantify the information covered in $\mathcal{T}_i$ and $\theta_0^{(k)}$, we here propose two representative factors: 1) the loss $\mathcal{L}(\mathcal{D}_i^q; \theta_i^{(k)})$ on the query set, where $\theta_i^{(k)}$ is updated by performing a few-gradient steps starting from $\theta_0^{(k)}$; 2) the gradient similarity between the support and target sets with respect to the current meta-model $\theta_0^{(k)}$, i.e., $\left\langle \nabla_{\theta_0^{(k)}} \mathcal{L}(\mathcal{D}_i^s; \theta_0^{(k)}), \nabla_{\theta_0^{(k)}} \mathcal{L}(\mathcal{D}_i^q; \theta_0^{(k)}) \right\rangle$. Here, we use inner product as an exemplary similarity measurement, other metrics like cosine similarity can also be applied in practice. They are associated with the learning outcome and learning process of the task $\mathcal{T}_i$, respectively. Specifically, the gradient similarity signifies the generalization gap from the support to the query set. A large query loss may represent a true hard task if the gradient similarity is large; a task with noises in its query set, however, could lead to a large query loss but small gradient similarity. Considering these two factors simultaneously, we reformulate Eqn. (2) as:

$$w_i^{(k)} = g\left( \mathcal{L}(\mathcal{D}_i^q; \theta_i^{(k)}), \left\langle \nabla_{\theta_0^{(k)}} \mathcal{L}(\mathcal{D}_i^s; \theta_0^{(k)}), \nabla_{\theta_0^{(k)}} \mathcal{L}(\mathcal{D}_i^q; \theta_0^{(k)}) \right\rangle; \phi^{(k)} \right). \tag{3}$$

After obtaining the sampling probability weight $w_i^{(k)}$, we directly use it to sample $B$ tasks from the candidate task pool for the current meta-training iteration, where a larger value of $w_i^{(k)}$ represents higher probability. The out-loop optimization is revised as:

$$\theta_0^{(k+1)} = \theta_0^{(k)} - \beta \frac{1}{B} \sum_{i=1}^{B} \mathcal{L}(\mathcal{D}_i^q; \theta_i^{(k)}). \tag{4}$$

### 4.2 Bi-level Optimization with Gradient Approximation

In this subsection, we aim to discuss how to jointly learn the parameters of neural scheduler $\phi$ and the meta-model $\theta_0$ during the meta-training process. Instead of directly using the meta-training tasks to optimize both meta-model and the neural scheduler, ATS aims to optimize the loss on a disparate validation set with $N_v$ tasks, i.e., $\{\mathcal{T}_v\}_{v=1}^{N_v}$, where the performance on the validation set could be regarded as the reward or fitness. Specifically, ATS searches the optimal parameter $\phi^*$ of the neural scheduler by minimizing the average loss $\frac{1}{N_v} \sum_{v=1}^{N_v} \mathcal{L}_{val}(\mathcal{T}_v; \theta_0^*(\phi))$ over the validation tasks, where the optimal parameter $\theta_0^*$ is obtained by optimizing the meta-training loss $\frac{1}{B} \sum_{i=1}^{B} \mathcal{L}_{tr}(\mathcal{T}_i; \theta_0, \phi)$ over the sampled tasks. Formally, the bi-level optimization process is formulated as:

$$\min_{\phi} \frac{1}{N_v} \sum_{v=1}^{N_v} \mathcal{L}_{val}(\mathcal{T}_v; \theta_0^*(\phi)), \text{ where } \theta_0^*(\phi) = \arg\min_{\theta_0} \frac{1}{B} \sum_{i=1}^{B} \mathcal{L}_{tr}(\mathcal{T}_i; \theta_0, \phi) \tag{5}$$

It is computational expensive to optimize the inner loop in Eqn. (5) directly. Inspired by the differentiate hyperparameter optimization [18], we propose a strategy to approximate $\theta_0^*(\phi)$ by performing one gradient step starting from the current meta-model $\theta_0^{(k)}$ as:

$$\theta_0^*(\phi^{(k)}) \approx \tilde{\theta}_0^{(k+1)}(\phi^{(k)}) = \theta_0^{(k)} - \beta \nabla_{\theta_0^{(k)}} \frac{1}{B} \sum_{i=1}^{B} \mathcal{L}_{tr}(\mathcal{T}_i; \theta_i^{(k)}, \phi^{(k)}).$$

$$\text{s.t. } \theta_i^{(k)} = \theta_0^{(k)} - \alpha \nabla_{\theta} \mathcal{L}(\mathcal{D}_i^s; \theta), \tag{6}$$

where the task-specific parameter $\theta_i^{(k)}$ is adapted with a few gradient steps starting from $\theta_0^{(k)}$. As such, for each meta-training iteration, the bi-level optimization process in Eqn. (5) is revised as:

$$\phi^{(k+1)} \leftarrow \min_{\phi^{(k)}} \frac{1}{N_v} \sum_{v=1}^{N_v} \mathcal{L}_{val}(\mathcal{T}_v; \tilde{\theta}_0^{(k+1)}(\phi^{(k)})),$$

$$\text{s.t., } \tilde{\theta}_0^{(k+1)}(\phi^{(k)}) = \theta_0^{(k)} - \beta \nabla_{\theta_0^{(k)}} \frac{1}{B} \sum_{i=1}^{B} \mathcal{L}_{tr}(\mathcal{T}_i; \theta_0^{(k)}, \phi^{(k)}). \tag{7}$$

We then focus on optimizing the neural scheduler in the outer loop of Eqn. (7). In ATS, tasks are sampled from the candidate task pool according to the sampling probabilities $w_i^{(k)}$. It is intractable to directly optimizing the validation loss $\mathcal{L}_{val}$, since the sampling process is non-differentiable. We thereby use a policy gradient method to optimize $\phi^{(k)}$, where REINFORCE [29] is adopted. We also

---

**Algorithm 1** Meta-training Process with ATS

---

**Require:** learning rates $\alpha$, $\beta$, task distribution $p(\mathcal{T})$, batch size $B$, candidate task pool size $N^{pool}$
1: Initialize the meta-model $\theta_0$ and the neural scheduler $\phi$
2: **for** each training iteration $k$ **do**
3:   Randomly select $N^{pool}$ tasks and construct the candidate task pool
4:   **for** each task $\mathcal{T}_i$ in the candidate task pool **do**
5:     Compute two factors $\mathcal{L}(\mathcal{D}_i^q; \theta_i^{(k)})$ and $\left\langle \nabla_{\theta_0^{(k)}} \mathcal{L}(\mathcal{D}_i^s; \theta_0^{(k)}), \nabla_{\theta_0^{(k)}} \mathcal{L}(\mathcal{D}_i^q; \theta_0^{(k)}) \right\rangle$ by using the
       support set $\mathcal{D}_i^s$ and the query set $\mathcal{D}_i^q$
6:     Calculate the sampling probability $w_i^{(k)}$ by Eqn. (2)
7:   **end for**
8:   Sample $B$ tasks from the candidate task pool via the sampling probabilities $\mathbf{W}^{(k)}$
9:   Calculate the training loss and obtain a temporal meta-model via 1-step gradient descent
10:  Sample $N_v$ validation tasks
11:  Calculate the accuracy (reward) $R_i^{(k)}$ using the temporal meta-model $\tilde{\theta}_0^{(k+1)}$
12:  Update the neural scheduler by Eqn. (8) and get $\phi^{(k+1)}$
13:  Sample another $B$ tasks $\{\mathcal{T}_i^{'}\}_{i=1}^B$ by using the updated task scheduler $\phi^{(k+1)}$
14:  Update the meta-model as: $\theta_0^{(k+1)}(\phi^{(k)}) = \theta_0^{(k)} - \beta \nabla_{\theta_0^{(k)}} \frac{1}{B} \sum_{i=1}^B \mathcal{L}_{tr}(\mathcal{T}_i^{'}; \theta_0^{(k)}, \phi^{(k+1)})$
15: **end for**

---

equip a baseline function for REINFORCE to reduce the gradient variance. Regarding the accuracy of each sampled validation task $\mathcal{T}_i$ as the reward $R_i^{(k)}$, we define the optimization process as:

$$\phi^{(k+1)} \leftarrow \phi^{(k)} - \gamma \nabla_{\phi^{(k)}} \log P(\mathbf{W}^{(k)}; \phi^{(k)})(\frac{1}{N_v} \sum_{i=1}^{N_v} R_i^{(k)} - b), \tag{8}$$

where $\mathbf{W}^{(k)} = \{w_i^{(k)}\}_{i=1}^B$ and the baseline $b$ is defined as the moving average of validation accuracies. After updating the parameter of neural scheduler $\phi$, we use it to resample $B$ tasks from the candidate task pool and update the meta-model from $\theta_0^{(k)}$ to $\theta_0^{(k+1)}$ via Eqn. (7). The whole optimization algorithm of ATS is illustrated in Alg. 1.

## 5    Theoretical Analysis

In this section, we would extend the theoretical analysis in [6] to our problem of task scheduling, to theoretically study how the neural scheduler $g$ improves the meta-training loss as well as the optimization landscape. Without loss of generality, here we consider a weighted version of ATS with hard sampling, where the meta-model $\theta_0$ is updated by solving the meta-training loss weighted by the task sampling probability $w_i = g(\mathcal{T}_i, \theta_0; \phi)$ over all candidate tasks in the task pool, i.e.,

$$\theta_0^* = \arg\min_{\theta_0} \sum_{i=1}^{N^{pool}} w_i \mathcal{L}(\mathcal{D}_i^q; \theta_i), \quad \theta_i = \theta_0 - \alpha \nabla_\theta \mathcal{L}(\mathcal{D}_i^s; \theta). \tag{9}$$

Denote the meta-training loss without and with the task scheduler as $\mathcal{L}(\theta_0) = \frac{1}{N^{pool}} \sum_{i=1}^{N^{pool}} \mathcal{L}(\mathcal{D}_i^q; \theta_i)$ and $\mathcal{L}^w(\theta_0) = \sum_{i=1}^{N^{pool}} w_i \mathcal{L}(\mathcal{D}_i^q; \theta_i)$, respectively. Then we have the following result:

**Proposition 1.** *Suppose that* $\mathbf{w} = [w_1, \cdots, w_{N^{pool}}]$ *denotes the random variable for sampling probabilities,* $\boldsymbol{\mathcal{L}}_{\theta_0} = [\mathcal{L}(\mathcal{D}_1^q; \theta_0), \cdots, \mathcal{L}(\mathcal{D}_{N^{pool}}^q; \theta_0)]$ *denotes the random variable for the loss using the meta-model, and* $\boldsymbol{\nabla}_{\theta_0} = [\langle \nabla_{\theta_0} \mathcal{L}(\mathcal{D}_1^s; \theta_0), \nabla_{\theta_0} \mathcal{L}(\mathcal{D}_1^q; \theta_0) \rangle, \cdots, \langle \nabla_{\theta_0} \mathcal{L}(\mathcal{D}_{N^{pool}}^s; \theta_0), \nabla_{\theta_0} \mathcal{L}(\mathcal{D}_{N^{pool}}^q; \theta_0) \rangle]$ *denotes the random variable for the inner product between gradients of the support and query sets with respect to the meta-model. Then the following equation connecting the meta-learning losses with and without the task scheduler holds:*

$$\mathcal{L}^w(\theta_0) = \mathcal{L}(\theta_0) + \mathrm{Cov}(\boldsymbol{\mathcal{L}}_{\theta_0}, \mathbf{w}) - \alpha \mathrm{Cov}(\boldsymbol{\nabla}_{\theta_0}, \mathbf{w}). \tag{10}$$

From Proposition 1, we conclude that the task scheduler improves the meta-training loss, as long as the sampling probability $\mathbf{w}$ is negatively correlated with the loss but positively correlated with the gradient similarity between the support and the query set. Specifically, if the loss $\mathcal{L}(\mathcal{D}_i^q; \theta_i)$ is large

as a result of a quite challenging or noisy task $\mathcal{T}_i$, the sampling probability $w_i$ is expected to be small. Moreover, a large value of $w_i$ is anticipated, when a large inner product between the gradients of the support and the query set with respect to the meta-model $\langle\nabla_{\theta_0}\mathcal{L}(\mathcal{D}_i^s;\theta_0),\nabla_{\theta_0}\mathcal{L}(\mathcal{D}_i^{\tilde{q}};\theta_0)\rangle$ signifies that the generalization gap from the support set $\mathcal{D}_i^s$ to the query set $\mathcal{D}_i^q$ is small.

Consistent with [6], the optimal meta-model is assumed to also minimize the covariance $\mathrm{Cov}(\mathcal{L}_{\theta_0},\mathbf{w})$ and maximize the covariance $\mathrm{Cov}(\nabla_{\theta_0},\mathbf{w})$, i.e., $\theta_0^* = \arg\min\mathcal{L}(\theta_0) = \arg\min[\mathrm{Cov}(\mathcal{L}_{\theta_0},\mathbf{w}) - \alpha\mathrm{Cov}(\nabla_{\theta_0},\mathbf{w})]$. Under this assumption, the task scheduler does not change the global minimum, i.e., $\theta_0^* = \arg\min\mathcal{L}(\theta_0) = \arg\min\mathcal{L}^w(\theta_0)$, while modifies the optimization landscape as the following.

**Proposition 2.** *With the sampling probability defined as*

$$w_i^* = \frac{e^{-\left[\mathcal{L}(\mathcal{D}_i^q;\theta_0^*)-\alpha\left\langle\nabla_{\theta_0}\mathcal{L}(\mathcal{D}_i^s;\theta_0^*),\nabla_{\theta_0}\mathcal{L}(\mathcal{D}_i^q;\theta_0^*)\right\rangle\right]}}{\sum_{i=1}^B e^{-\left[\mathcal{L}(\mathcal{D}_i^q;\theta_0^*)-\alpha\left\langle\nabla_{\theta_0}\mathcal{L}(\mathcal{D}_i^s;\theta_0^*),\nabla_{\theta_0}\mathcal{L}(\mathcal{D}_i^q;\theta_0^*)\right\rangle\right]}}, \tag{11}$$

*the following hold:*

$$\forall\theta_0: \mathrm{Cov}(\mathcal{L}_{\theta_0}-\alpha\nabla_{\theta_0},e^{-(\mathcal{L}_{\theta_0^*}-\alpha\nabla_{\theta_0^*})}) \geq 0, \qquad\qquad \mathcal{L}^w(\theta_0)-\mathcal{L}^w(\theta_0^*) \geq \mathcal{L}(\theta_0)-\mathcal{L}(\theta_0^*),$$

$$\forall\theta_0: \mathrm{Cov}(\mathcal{L}_{\theta_0}-\alpha\nabla_{\theta_0},e^{-(\mathcal{L}_{\theta_0^*}-\alpha\nabla_{\theta_0^*})}) \leq -\mathrm{Var}(\mathcal{L}_{\theta_0^*}-\alpha\nabla_{\theta_0^*}), \quad \mathcal{L}^w(\theta_0)-\mathcal{L}^w(\theta_0^*) \leq \mathcal{L}(\theta_0)-\mathcal{L}(\theta_0^*).$$

Proposition 2 sheds light on how the optimization landscape is improved by an ideal task scheduler: 1) for those parameters $\theta_0$ that are far from the optimal meta-model $\theta_0^*$ (i.e., $\mathrm{Cov}(\mathcal{L}_{\theta_0}-\alpha\nabla_{\theta_0},e^{-(\mathcal{L}_{\theta_0^*}-\alpha\nabla_{\theta_0^*})}) \geq 0$), the gradients towards the direction of $\theta_0^*$ become overall steeper for speed-up; 2) for those parameters $\theta_0$ that are within the variance of the optimum (i.e., $\mathrm{Cov}(\mathcal{L}_{\theta_0}-\alpha\nabla_{\theta_0},e^{-(\mathcal{L}_{\theta_0^*}-\alpha\nabla_{\theta_0^*})}) \leq -\mathrm{Var}(\mathcal{L}_{\theta_0^*}-\alpha\nabla_{\theta_0^*}))$, the minima tends to be flat with better generalization ability [7, 14]. Though the optimal meta-model $\theta_0^*$ remains unknown and the ideal task scheduler with the sampling probabilities in (11) is inaccessible, we learn a neural scheduler to dynamically accommodate the changes of both the loss $\mathcal{L}_{\theta_0}$ and the gradient similarity $\nabla_{\theta_0}$. Detailed proofs of Proposition 1 and 2 are provided in Appendix A.

## 6 Experiments

In this section, we empirically demonstrate the effectiveness of the proposed ATS through comprehensive experiments on both regression and classification problems. Specifically, two challenging settings are studied: meta-learning with noise and limited budgets.

**Dataset Description and Model Structure.** We conduct comprehensive experiments on two datasets. First, we use miniImagenet as the classification dataset, where we apply the conventional N-way, K-shot setting to create tasks [4]. For miniImagenet, we use the standard model with four convolutional blocks, where each block contains 32 neurons. We report accuracy with 95% confidence interval over all meta-testing tasks. The second dataset aims to predict the activity of drug compounds [21], where each task as an assay covers drug compounds for one target protein. There are 4,276 assays in total, and we split 4,100 / 76 / 100 tasks for meta-training / validation / testing, respectively. We use two fully connected layers in the drug activity prediction as the base model, where each layer contains 500 neurons. For each assay, the performance is measured by the square of Pearson coefficient ($R^2$) between the predicted values and the actual values. Follow [21], we report the mean and medium $R^2$ as well as the number of assays with $R^2 > 0.3$, all of which are considered as reliable metrics in the pharmacology domain. We provide more detailed descriptions of datasets in Appendix B.1.

In terms of the neural scheduler $\phi$, we separately encode the loss on the query set and the gradient similarity by two bi-directional LSTM network. The percentage of training iterations are also fed into the neural scheduler to indicate the progress of meta-training. Finally, all encoded information are concatenated and feed into a two-layer MLP for predicting the sampling probability (More detail descriptions are provided in Appendix B.2).

**Baselines.** We compare the proposed ATS with the following two categories of baselines. The first category contains easy-to-implement example sampling methods that can be adapted for task scheduling, including focal loss (FocalLoss) [16] and self-paced learning loss (SPL) [13]. The second category covers state-of-the-art task schedulers for meta-learning that are non-adaptive, which includes DAML [15], GCP [17], and PAML [11]. Note that GCP is a class-driven task scheduler

Table 1: Overall performance on meta-learning with noise. For miniImagenet, we report the average accuracy with 95% confidence interval. For drug activity prediction, the performance is evaluated by mean $R^2$, medium $R^2$, and the number of assays with $R^2 > 0$.

| Model | miniImagenet-noisy | | Drug-noisy | | |
| | 5-way 1-shot | 5-way 5-shot | mean | medium | >0.3 |
| --- | --- | --- | --- | --- | --- |
| Uniform | $41.67 \pm 0.80\%$ | $55.80 \pm 0.71\%$ | 0.202 | 0.113 | 21 |
| SPL | $42.13 \pm 0.79\%$ | $56.19 \pm 0.70\%$ | 0.211 | 0.138 | 24 |
| FocalLoss | $41.91 \pm 0.78\%$ | $53.58 \pm 0.75\%$ | 0.205 | 0.106 | 23 |
| GCP | $41.86 \pm 0.75\%$ | $54.63 \pm 0.72\%$ | N/A | N/A | N/A |
| PAML | $41.49 \pm 0.74\%$ | $52.45 \pm 0.69\%$ | 0.204 | 0.120 | 24 |
| DAML | $41.26 \pm 0.73\%$ | $55.46 \pm 0.70\%$ | 0.197 | 0.113 | 24 |
| **ATS (Ours)** | $\mathbf{44.21 \pm 0.76\%}$ | $\mathbf{59.50 \pm 0.71\%}$ | $\mathbf{0.233}^*$ | $\mathbf{0.152}^*$ | $\mathbf{31}^*$ |

\* means the result are significant according to Student's T-test at level 0.01 compared to SPL

and thus it can not be applied to regression problems such as drug activity prediction here. We also slightly revise the loss of DAML so that it can be applied to both classification and regression problems. For all baselines and ATS, we use the same base model and adopt ANIL [22] as the backbone meta-learning algorithm.

## 6.1 Meta-learning with Noise

**Experimental Setup.** We first apply ATS on meta-learning with noisy tasks, where each noisy task is constructed by only adding noises on the labels of the support set. Therefore, each noisy task contains a noisy support set and a clean query set. In this way, adapting the meta-model on the support set causes inaccurate task-specific parameters and leads to negative impacts on the meta-training process. Specifically, for miniImagenet, we apply the symmetry flipping on the labels of the support set [27]. The default ratio of noisy tasks is set as 0.6. For the drug activity prediction, we sample the label noise $\epsilon$ from a Gaussian distribution $\eta * \mathcal{N}(0, 1)$, where the noise scalar $\eta$ is used to control the noise level. Note that, empirically we find that the effect of adding noise on the drug activity prediction is not as great as adding noise on the miniImagenet, and thus we add noise to all assays and use the scalar $\eta$ to control the noise ratio. By default, we set the noise scalar as 4 during the meta-training process. Besides, since ATS uses the clean validation task, for fair comparison, all other baselines are also fine-tuned on the validation tasks. Detailed experimental setups and hyperparameters are provided in Appendix C.

**Results.** Table 1 reports the overall results of ATS and other baselines. Our key observations are: (1) The performance of non-adaptive task schedulers (i.e., GCP, PAML, DAML) is similar to the uniform sampling, indicating that manually designing the task scheduler may not explain the complex dynamics of meta-learning, being sub-optimal. (2) ATS outperforms traditional example sampling methods and non-adaptive task schedulers, demonstrating its effectiveness on improving the robustness of meta-learning algorithms by adaptively adjusting the scheduler policy based on the real-time feedback of meta-model-related factors. The findings are further strengthened by the distribution comparison of sampling weights between clean and noisy tasks in Figure 2, where ATS pushes most noisy tasks to small weights (i.e., less contributions in meta-training).

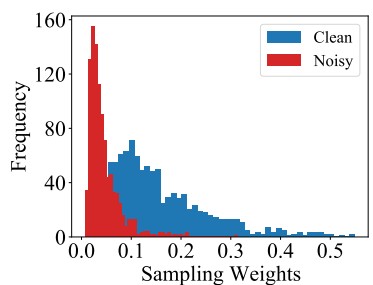

Figure 2: Distribution comparison of sampling weights between clean and noisy tasks.

**Ablation Study.** We further conduct an ablation study under the noisy task setting and investigate five ablation models detailed as follows.

- *Rank by Sim/Loss*: In the first ablation model, we heuristically determine and select tasks by ranking tasks according to a simple combination of the loss and the gradient similarity, i.e., Sim/Loss. We assume that tasks with higher gradient similarity but smaller losses should be prioritized.

- *Random $\phi$*: In Random $\phi$, we remove both meta-model-related factors, where the model parameter $\phi$ of the neural scheduler is randomly initialized at each meta-training iteration.

Table 2: Ablation Study under the meta-learning with noise setting.

| Ablation Model | miniImagenet-noisy | | Drug-noisy | | |
| | 5-way 1-shot | 5-way 5-shot | mean | medium | >0.3 |
| --- | --- | --- | --- | --- | --- |
| Random $\phi$ | $41.95 \pm 0.80\%$ | $56.07 \pm 0.71\%$ | 0.204 | 0.100 | 22 |
| Rank by Sim/Loss | $42.84 \pm 0.76\%$ | $57.90 \pm 0.68\%$ | 0.181 | 0.109 | 22 |
| $\phi$+Loss | $42.45 \pm 0.80\%$ | $56.65 \pm 0.75\%$ | 0.212 | 0.122 | 27 |
| $\phi$+Sim | $42.28 \pm 0.82\%$ | $56.71 \pm 0.72\%$ | 0.214 | 0.122 | 29 |
| Reweighting | $42.19 \pm 0.80\%$ | $56.48 \pm 0.72\%$ | 0.217 | 0.118 | 28 |
| ATS ($\phi$+Loss+Sim) | $\mathbf{44.21 \pm 0.76\%}$ | $\mathbf{59.50 \pm 0.71\%}$ | $\mathbf{0.233^*}$ | $\mathbf{0.152^*}$ | $\mathbf{31^*}$ |

* means the result is significant according to Student's T-test at level 0.01 compared to Weighting

Table 3: Performance w.r.t. Noise Ratio. Under the miniImagenet 1-shot setting (Image), the noise ratio is controlled by the proportion of noisy tasks. In drug activity prediction, the noise ratio is determined by the value of noise scaler $\eta$. BNS represents the best non-adaptive scheduler.

| | Noise Ratio | 0.2 | 0.4 | 0.6 | 0.8 |
| --- | --- | --- | --- | --- | --- |
| Image | Uniform | $43.46 \pm 0.82\%$ | $42.92 \pm 0.78\%$ | $41.67 \pm 0.80\%$ | $36.53 \pm 0.73\%$ |
| | BNS | $44.04 \pm 0.81\%$ | $43.36 \pm 0.75\%$ | $42.13 \pm 0.79\%$ | $38.21 \pm 0.75\%$ |
| | **ATS (Ours)** | $\mathbf{45.55 \pm 0.80\%}$ | $\mathbf{44.50 \pm 0.86\%}$ | $\mathbf{44.21 \pm 0.76\%}$ | $\mathbf{42.18 \pm 0.73\%}$ |

| | Noise Scaler | $\eta$=2 | | | $\eta$=4 | | | $\eta$=6 | | | $\eta$=8 | | |
| --- | --- | --- | --- | --- | --- | --- | --- | --- | --- | --- | --- | --- | --- |
| Drug | Uniform | 0.222 | 0.139 | 26 | 0.202 | 0.113 | 21 | 0.196 | 0.131 | 22 | 0.194 | 0.100 | 21 |
| | BNS | 0.229 | 0.136 | 31 | 0.211 | 0.138 | 24 | 0.208 | 0.116 | 24 | 0.200 | 0.101 | 24 |
| | **ATS* (Ours)** | **0.235** | **0.160** | **33** | **0.233** | **0.152** | **31** | **0.221** | **0.136** | **28** | **0.219** | **0.133** | **28** |

* means all results are significant according to Student's T-test at level 0.01 compared to BNS

- *$\phi$+Loss or $\phi$+Sim*: The third ($\phi$+Loss) and forth ($\phi$+Sim) ablation models remove the gradient similarity between the support and query sets, as well as the loss of the query set, respectively.

- *Reweighting*: Instead of selecting tasks from the candidate pool, in the last ablation model, we direct reweigh all tasks in a meta batch, where the weights are learned via the neural scheduler.

We list the results of all ablation models in Table 2, where ATS ($\phi$+Loss+Sim) is also reported for comparison. The results indicate that (1) simply selecting tasks according to the ratio Sim/Loss significantly underperforms ATS since the contribution of each metric is hard to manually define. Besides, the contribution of each metric evolves as training proceeds, which has been ignored in such a simple combination but modeled in the neural scheduler with the percentage of training iterations as input; (2) the superiority of $\phi$+Loss and $\phi$+Sim over Random $\phi$ shows the effectiveness of both the query loss and the gradient similarity; (3) the performance gap between Reweight+Loss+Sim and ATS is potentially caused by the number of effective tasks, where more candidate tasks are considered by ATS; (4) including all meta-model-related factors (i.e., ATS) achieves the best performance, coinciding with our theoretical findings in Section 5.

**Effect of Noise Ratio.** We analyze the performance of ATS with respect to the noise ratio and show the results of miniImagenet and drug activity prediction in Table 3. The performances of the uniform sampling and the best non-adaptive scheduler (BNS) are also reported for comparison. We summarize the key findings: (1) ATS consistently outperforms the uniform sampling and the best non-adaptive task scheduler, indicating its effectiveness of adaptively sampling tasks to guide the meta-training process; (2) With the increase of the noise ratio, ATS achieves more significant improvements. In particular, the results of ATS in miniImagenet are almost stable even with a very large noise ratio, suggesting that involving an adaptive task scheduler does improve the robustness of the model.

**Analysis of the Meta-model-related Factors.** To analyze our motivations about designing the meta-model-related factors, we randomly select 1000 meta-training tasks and visualize the correlation between the sampling weight $w_i^k$ and each factor in Figures 3a-3d. In these figures, we rank the sampling weight $w_i^k$ and normalize the ranking to $[0, 1]$, where a larger normalized ranking value is associated with a larger sampling weight. Then, we split these tasks into 20 bins according to the rank of sampling weights. For tasks within each bin, we show the mean and standard error of their query losses and gradient similarities. According to these figures, we find that tasks with larger losses are

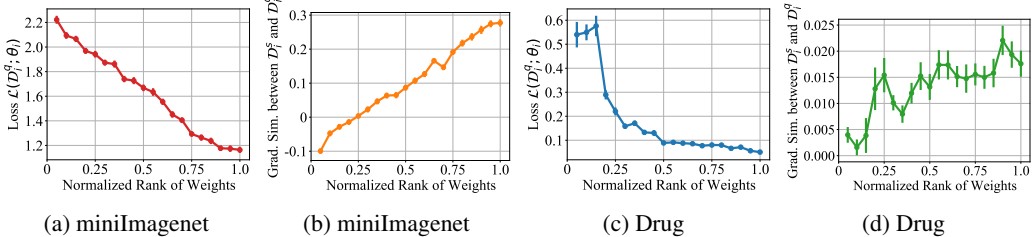

| (a) miniImagenet | (b) miniImagenet | (c) Drug | (d) Drug |

Figure 3: Correlation between sampling weight $w_i$ and (a)&(c) query set loss $\mathcal{L}(\mathcal{D}_i^q; \theta_i)$; (b)&(d): gradient similarity between $\mathcal{D}_i^s$ and $\mathcal{D}_i^q$ under the meta-learning with noise setting. The larger normalized rank of weights correspond to larger sampling weights.

Table 4: Overall performance on meta-learning with limited budgets. For miniImagenet, we control the number of meta-training classes. For drug activity prediction, all meta-training tasks are used for meta-training under this setting.

| Model | miniImagenet-Limited | | Drug-Full | | |
| | 5-way 1-shot | 5-way 5-shot | mean | medium | >0.3 |
| --- | --- | --- | --- | --- | --- |
| Uniform | $33.61 \pm 0.66\%$ | $45.97 \pm 0.65\%$ | 0.233 | 0.140 | 33 |
| SPL | $34.28 \pm 0.65\%$ | $46.05 \pm 0.69\%$ | 0.232 | 0.135 | 29 |
| FocalLoss | $33.11 \pm 0.65\%$ | $46.12 \pm 0.70\%$ | 0.229 | 0.140 | 28 |
| GCP | $34.69 \pm 0.67\%$ | $46.86 \pm 0.68\%$ | N/A | N/A | N/A |
| PAML | $33.64 \pm 0.62\%$ | $45.01 \pm 0.69\%$ | 0.238 | 0.144 | 32 |
| DAML | $34.83 \pm 0.69\%$ | $46.66 \pm 0.67\%$ | 0.227 | 0.141 | 28 |
| **ATS (Ours)** | $\mathbf{35.15 \pm 0.67\%}$ | $\mathbf{47.76 \pm 0.68\%}$ | $\mathbf{0.252}^*$ | $\mathbf{0.179}^*$ | $\mathbf{36}^*$ |

\* means the result is significant according to Student's T-test at level 0.01 compared to PAML

associated with smaller sampling weights, verifying our assumption that noisy tasks (large query loss) tend to have smaller sampling weight (Figures 3a, 3c). Our motivation is further strengthened by the fact that tasks with more similar support and query sets have larger sampling weights (Figures 3b, 3d).

## 6.2 Meta-learning with Limited Budgets

**Experimental Setup.** We further analyze the effectiveness of ATS under the meta-learning setting with limited budgets. Follow [4], In few-shot classification problem, each training episode is a few-shot task by subsampling classes as well as data points and two episodes that share the same classes are considered to be the same task. Thus, we treat the budgets in meta-learning as the number of meta-training tasks. In miniImagenet, the original number of meta-training classes is 64, corresponding to more than 7 million 5-way combinations. Thus, we control the budgets by reducing the number of meta-training classes to 16, resulting in 4,368 combinations. For drug activity prediction, since it only has 4,100 tasks, we do not reduce the number of tasks and use the full dataset for meta-training. We provide more discussions about the setup in Appendix D.

**Results.** We report the performance on miniImagenet and drug activity prediction in Table 4. Aligning with the meta-learning with noise setting, ATS consistently outperforms other baselines with limited budgets. Besides, compared with the results in miniImagenet, ATS achieves more significant improvement in the drug activity prediction problem under this setting. This is what we expected – as we have discussed in Section 1, the drug dataset contains noisy and imbalanced tasks.

The effectiveness of ATS is further strengthened by the ablation study under the limited budgets setting, where we report the results in Table 5. Similar to the findings in the noisy task setting, the ablation study further verifies the contributions of the proposed two meta-model-related factors on the meta-training process.

**Analysis of the Meta-model-related Factors.** Under the limited budgets setting, we analyze the correlation between the sampling weight and the meta-model-related factors in Figure 4a-4d. From these figures, we can see that the gradient similarity indeed reflects the task difficulty (Figure 4b, 4d), where larger similarities correspond to more useful tasks (i.e., larger weights). Interestingly, the

Table 5: Performance (accuracy ± 95% confidence interval) of different ablated versions of ATS under the setting of meta-learning with limited tasks.

| Ablation Model | miniImagenet-Limited | | Drug-Full | | |
| | 5-way 1-shot | 5-way 5-shot | mean | medium | >0.3 |
| --- | --- | --- | --- | --- | --- |
| Random $\phi$ | 33.97 ± 0.63% | 46.37 ± 0.70% | 0.238 | 0.159 | 35 |
| Rank by Sim/Loss | 33.42 ± 0.64% | 46.38 ± 0.70% | 0.187 | 0.099 | 24 |
| $\phi$+Loss | 34.08 ± 0.66% | 46.48 ± 0.67% | 0.241 | 0.171 | 36 |
| $\phi$+Sim | 34.46 ± 0.65% | 47.34 ± 0.70% | 0.246 | 0.161 | 34 |
| Reweighting | 35.03 ± 0.65% | 46.70 ± 0.65% | 0.248 | 0.158 | 32 |
| ATS ($\phi$+Loss+Sim) | **35.15 ± 0.67%** | **47.76 ± 0.68%** | **0.252** | **0.179** | **36** |

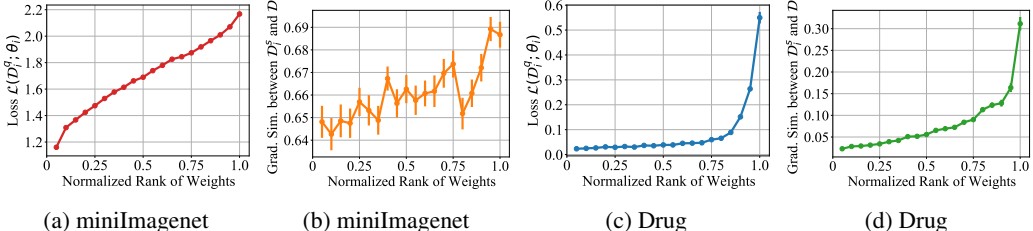

(a) miniImagenet     (b) miniImagenet     (c) Drug     (d) Drug

Figure 4: Correlation between weight $w_i$ and (a)&(c) query loss; (b)&(d): gradient similarity. Larger normalized rank of weights correspond to larger sampling weights.

correlation between the query loss and the sampling weight under the limited budgets setting is opposite to the correlation under the noisy setting. It is expected since tasks with larger query losses are associated with "more difficult" tasks under this setting, which may contain more valuable information that further benefits the meta-training process.

**Effect of the Budgets.** In addition, we analyze the effect of budgets by changing the number of meta-training tasks in miniImagenet. The results of uniform sampling, GCP (best non-adaptive task scheduler), ATS under 1-shot scenario are illustrated in Table 6. We observe that our model achieves the best performance in all scenarios. In addition, compared with uniform sampling, ATS achieves more significant improvements with less budgets, indicating its effectiveness on improving the meta-training efficiency.

Table 6: Performance w.r.t. budgets (the number of meta-training classes). Accuracy ± 95% confidence interval is reported.

| Budgets | 16 | 32 | 48 | 64 |
| --- | --- | --- | --- | --- |
| Uniform | 33.61 ± 0.66% | 40.48 ± 0.75% | 44.07 ± 0.80% | 45.73 ± 0.79% |
| GCP | 34.69 ± 0.67% | 41.27 ± 0.74% | 44.30 ± 0.79% | 45.35 ± 0.81% |
| **ATS (Ours)** | **35.15 ± 0.67%** | **41.68 ± 0.78%** | **44.89 ± 0.79%** | **46.27 ± 0.80%** |

# 7 Conclusion and Discussion

This paper proposes a new adaptive task sampling strategy (ATS) to improve the meta-training process. Specifically, we design a neural scheduler with two meta-model-related factors. At each meta-training iteration, the neural scheduler predicts the probability of each meta-training task being sampled according to the received factors from each candidate task. The meta-model and the neural scheduler are optimized in a bi-level optimization framework. Our experiments demonstrate the effectiveness of the proposed ATS under the settings of meta-learning with noise and limited budgets.

One limitation in this paper is that we only consider how to adaptively schedule tasks during the meta-training process. In the future, it would be meaningful to investigate how to incorporate the task scheduler with the sample scheduler within each task. Another limitation is that using ATS is more computationally expensive than random sampling since we alternatively learn the neural scheduler and the meta-model. It would be interesting to explore how to reduce the computational cost, e.g., compressing the neural scheduler.

## Acknowledgement

This work was supported in part by JPMorgan Chase & Co. Any views or opinions expressed herein are solely those of the authors listed, and may differ from the views and opinions expressed by JPMorgan Chase & Co. or its affiliates. This material is not a product of the Research Department of J.P. Morgan Securities LLC. This material should not be construed as an individual recommendation for any particular client and is not intended as a recommendation of particular securities, financial instruments or strategies for a particular client. This material does not constitute a solicitation or offer in any jurisdiction. This work was also supported in part by the start-up grant from City University of Hong Kong (9610512). Besides, Y. Wei would like to acknowledge the support from the Tencent AI Lab Rhino-Bird Gift Fund. D. Lian was supported by grants from the National Natural Science Foundation of China # 62022077.

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
