# Meta-learning with an Adaptive Task Scheduler

**Huaxiu Yao**[1]*, **Yu Wang**[2], **Ying Wei**[3], **Peilin Zhao**[4]
**Mehrdad Mahdavi**[5], **Defu Lian**[2], **Chelsea Finn**[1]
[1]Stanford University, [2]University of Science and Technology, [3] Tencent AI Lab
[4]Pennsylvania State University, [5]City University of Hong Kong
[1]{huaxiu,cbfinn}@cs.stanford.edu, [2]{wangyu,liandefu}@ustc.edu.cn
[3]yingwei@cityu.edu.hk, [4]masonzhao@tencent.com, [5]mzm616@psu.edu

## A  Proof of Propositions

### A.1  Proof of Proposition 1

Recall that Eqn. (1) connecting the meta-learning losses without and with the task scheduler holds in Proposition 1, i.e.,

$$\mathcal{L}^w(\theta_0) = \mathcal{L}(\theta_0) + \mathrm{Cov}(\boldsymbol{\mathcal{L}}_{\theta_0}, \mathbf{w}) - \alpha \mathrm{Cov}(\boldsymbol{\nabla}_{\theta_0}, \mathbf{w}). \tag{1}$$

We provide the following proof for the Proposition 1 as:

*Proof.* We re-write the meta-training loss without the task scheduler as,

$$\mathcal{L}(\theta_0) = \frac{1}{N^{pool}} \sum_{i=1}^{N^{pool}} \mathcal{L}(\mathcal{D}_i^q; \theta_i)$$

$$= \frac{1}{N^{pool}} \sum_{i=1}^{N^{pool}} (\mathcal{L}(\mathcal{D}_i^q; \theta_i) + \mathcal{L}(\mathcal{D}_i^q; \theta_i) - \mathcal{L}(\mathcal{D}_i^q; \theta_i)) - \sum_{i=1}^{N^{pool}} w_i(\mathcal{L}(\mathcal{D}_i^q; \theta_i) - \mathcal{L}(\mathcal{D}_i^q; \theta_i))$$

$$= \mathcal{L}^w(\theta_0) - \sum_{i=1}^{N^{pool}} \left( w_i \mathcal{L}(\mathcal{D}_i^q; \theta_i) + w_i \left[ \frac{1}{N^{pool}} \sum_{i=1}^{N^{pool}} \mathcal{L}(\mathcal{D}_i^q; \theta_i) \right] + \mathcal{L}(\mathcal{D}_i^q; \theta_i) \left[ \frac{1}{N^{pool}} \sum_{i=1}^{N^{pool}} w_i \right] \right) \tag{2}$$

$$\quad - \sum_{i=1}^{N^{pool}} \left[ \frac{1}{N^{pool}} \sum_{i=1}^{N^{pool}} \mathcal{L}(\mathcal{D}_i^q; \theta_i) \right] \left[ \frac{1}{N^{pool}} \sum_{i=1}^{N^{pool}} w_i \right]$$

$$= \mathcal{L}^w(\theta_0) - \sum_{i=1}^{N^{pool}} \left[ \mathcal{L}(\mathcal{D}_i^q; \theta_i) - \frac{1}{N^{pool}} \sum_{i=1}^{N^{pool}} \mathcal{L}(\mathcal{D}_i^q; \theta_i) \right] \left[ w_i - \frac{1}{N^{pool}} \sum_{i=1}^{N^{pool}} w_i \right].$$

The third equation holds because the sum of sampling probabilities within the pool of all candidate tasks amounts to 1. We approximate the meta-training loss with regard to the $i$-th task, i.e., $\mathcal{L}(\mathcal{D}_i^q; \theta_i)$, with Taylor expansion, where we assume that it takes one step gradient descent from the meta-model $\theta_0$ to $\theta_i$. Thus,

$$\mathcal{L}(\mathcal{D}_i^q; \theta_i) = \mathcal{L}(\mathcal{D}_i^q; \theta_0 - \alpha \nabla_\theta \mathcal{L}(\mathcal{D}_i^s; \theta_0))$$

$$= \mathcal{L}(\mathcal{D}_i^q; \theta_0) - \alpha \langle \nabla_{\theta_0} \mathcal{L}(\mathcal{D}_i^s; \theta_0), \nabla_{\theta_0} \mathcal{L}(\mathcal{D}_i^q; \theta_0) \rangle + \alpha^2 \nabla_{\theta_0} \mathcal{L}(\mathcal{D}_i^s; \theta_0)^T \nabla_{\theta_0}^2 \mathcal{L}(\mathcal{D}_i^q; \theta_0) \nabla_{\theta_0} \mathcal{L}(\mathcal{D}_i^s; \theta_0)$$

$$\quad + O(\alpha^3 \|\nabla_{\theta_0} \mathcal{L}(\mathcal{D}_i^q; \theta_0)\|^3)$$

$$\approx \mathcal{L}(\mathcal{D}_i^q; \theta_0) - \alpha \langle \nabla_{\theta_0} \mathcal{L}(\mathcal{D}_i^s; \theta_0), \nabla_{\theta_0} \mathcal{L}(\mathcal{D}_i^q; \theta_0) \rangle. \tag{3}$$

---

*H. Yao and Y. Wang contribute equally; correspondence to: Y. Wei

35th Conference on Neural Information Processing Systems (NeurIPS 2021).

Substituting Eqn. (3) into Eqn. (2), we finish our proof:

$$\mathcal{L}(\theta_0) = \mathcal{L}^w(\theta_0) - \sum_{i=1}^{N^{pool}} \left[ \mathcal{L}(\mathcal{D}_i^q; \theta_0) - \frac{1}{N^{pool}} \sum_{i=1}^{N^{pool}} \mathcal{L}(\mathcal{D}_i^q; \theta_0) \right] \left[ w_i - \frac{1}{N^{pool}} \sum_{i=1}^{N^{pool}} w_i \right]$$

$$+ \alpha \sum_{i=1}^{N^{pool}} \left[ \langle \nabla_{\theta_0} \mathcal{L}(\mathcal{D}_i^s; \theta_0), \nabla_{\theta_0} \mathcal{L}(\mathcal{D}_i^q; \theta_0) \rangle - \frac{1}{N^{pool}} \sum_{i=1}^{N^{pool}} \langle \nabla_{\theta_0} \mathcal{L}(\mathcal{D}_i^s; \theta_0), \nabla_{\theta_0} \mathcal{L}(\mathcal{D}_i^q; \theta_0) \rangle \right] \left[ w_i - \frac{1}{N^{pool}} \sum_{i=1}^{N^{pool}} w_i \right]$$

$$= \mathcal{L}^w(\theta_0) - \mathrm{Cov}(\boldsymbol{\mathcal{L}}_{\theta_0}, \mathbf{w}) + \alpha \mathrm{Cov}(\boldsymbol{\nabla}_{\theta_0}, \mathbf{w}). \tag{4}$$

$\square$

### A.2  Proof of Proposition 2

Recall the Proposition 2 as: with the sampling probability defined as

$$w_i^* = \frac{e^{-\left[ \mathcal{L}(\mathcal{D}_i^q; \theta_0^*) - \alpha \langle \nabla_{\theta_0} \mathcal{L}(\mathcal{D}_i^s; \theta_0^*), \nabla_{\theta_0} \mathcal{L}(\mathcal{D}_i^q; \theta_0^*) \rangle \right]}}{\sum_{i=1}^{N^{pool}} e^{-\left[ \mathcal{L}(\mathcal{D}_i^q; \theta_0^*) - \alpha \langle \nabla_{\theta_0} \mathcal{L}(\mathcal{D}_i^s; \theta_0^*), \nabla_{\theta_0} \mathcal{L}(\mathcal{D}_i^q; \theta_0^*) \rangle \right]}}, \tag{5}$$

the followings hold:

$$\forall \theta_0 : \mathrm{Cov}(\boldsymbol{\mathcal{L}}_{\theta_0} - \alpha \boldsymbol{\nabla}_{\theta_0}, e^{-(\boldsymbol{\mathcal{L}}_{\theta_0^*} - \alpha \boldsymbol{\nabla}_{\theta_0^*})}) \geq 0, \qquad\qquad \mathcal{L}^w(\theta_0) - \mathcal{L}^w(\theta_0^*) \geq \mathcal{L}(\theta_0) - \mathcal{L}(\theta_0^*),$$

$$\forall \theta_0 : \mathrm{Cov}(\boldsymbol{\mathcal{L}}_{\theta_0} - \alpha \boldsymbol{\nabla}_{\theta_0}, e^{-(\boldsymbol{\mathcal{L}}_{\theta_0^*} - \alpha \boldsymbol{\nabla}_{\theta_0^*})}) \leq -\mathrm{Var}(\boldsymbol{\mathcal{L}}_{\theta_0^*} - \alpha \boldsymbol{\nabla}_{\theta_0^*}), \quad \mathcal{L}^w(\theta_0) - \mathcal{L}^w(\theta_0^*) \leq \mathcal{L}(\theta_0) - \mathcal{L}(\theta_0^*). \tag{6}$$

*Proof.* From Eqn. (1), we conclude that,

$$\mathcal{L}^w(\theta_0) - \mathcal{L}^w(\theta_0^*) = \mathcal{L}(\theta_0) + \mathrm{Cov}(\boldsymbol{\mathcal{L}}_{\theta_0}, \mathbf{w}) - \alpha \mathrm{Cov}(\boldsymbol{\nabla}_{\theta_0}, \mathbf{w})$$
$$- \mathcal{L}(\theta_0^*) - \mathrm{Cov}(\boldsymbol{\mathcal{L}}_{\theta_0^*}, \mathbf{w}) + \alpha \mathrm{Cov}(\boldsymbol{\nabla}_{\theta_0^*}, \mathbf{w})$$
$$= \mathcal{L}(\theta_0) - \mathcal{L}(\theta_0^*) + \mathrm{Cov}(\boldsymbol{\mathcal{L}}_{\theta_0} - \alpha \boldsymbol{\nabla}_{\theta_0}, \mathbf{w}) - \mathrm{Cov}(\boldsymbol{\mathcal{L}}_{\theta_0^*} - \alpha \boldsymbol{\nabla}_{\theta_0^*}, \mathbf{w}). \tag{7}$$

By substituting the sampling probability defined in (5) into (7) and defining $W = \sum_{i=1}^{N^{pool}} e^{-\left[ \mathcal{L}(\mathcal{D}_i^q; \theta_0^*) - \alpha \langle \nabla_{\theta_0} \mathcal{L}(\mathcal{D}_i^s; \theta_0^*), \nabla_{\theta_0} \mathcal{L}(\mathcal{D}_i^q; \theta_0^*) \rangle \right]}$, we have,

$$\mathcal{L}^w(\theta_0) - \mathcal{L}^w(\theta_0^*) = \mathcal{L}(\theta_0) - \mathcal{L}(\theta_0^*) + \frac{1}{W} \mathrm{Cov}(\boldsymbol{\mathcal{L}}_{\theta_0} - \alpha \boldsymbol{\nabla}_{\theta_0}, e^{-(\boldsymbol{\mathcal{L}}_{\theta_0^*} - \alpha \boldsymbol{\nabla}_{\theta_0^*})})$$
$$- \frac{1}{W} \mathrm{Cov}(\boldsymbol{\mathcal{L}}_{\theta_0^*} - \alpha \boldsymbol{\nabla}_{\theta_0^*}, e^{-(\boldsymbol{\mathcal{L}}_{\theta_0^*} - \alpha \boldsymbol{\nabla}_{\theta_0^*})}). \tag{8}$$

Meanwhile, the lower and upper bound of $\mathrm{Cov}(\boldsymbol{\mathcal{L}}_{\theta_0^*} - \alpha \boldsymbol{\nabla}_{\theta_0^*}, e^{-(\boldsymbol{\mathcal{L}}_{\theta_0^*} - \alpha \boldsymbol{\nabla}_{\theta_0^*})})$ are $\mathrm{Cov}(\boldsymbol{\mathcal{L}}_{\theta_0^*} - \alpha \boldsymbol{\nabla}_{\theta_0^*}, -(\boldsymbol{\mathcal{L}}_{\theta_0^*} - \alpha \boldsymbol{\nabla}_{\theta_0^*}))$ and 0, respectively, i.e.,

$$-\mathrm{Var}(\boldsymbol{\mathcal{L}}_{\theta_0^*} - \alpha \boldsymbol{\nabla}_{\theta_0^*}) = \mathrm{Cov}(\boldsymbol{\mathcal{L}}_{\theta_0^*} - \alpha \boldsymbol{\nabla}_{\theta_0^*}, -(\boldsymbol{\mathcal{L}}_{\theta_0^*} - \alpha \boldsymbol{\nabla}_{\theta_0^*})) \leq \mathrm{Cov}(\boldsymbol{\mathcal{L}}_{\theta_0^*} - \alpha \boldsymbol{\nabla}_{\theta_0^*}, e^{-(\boldsymbol{\mathcal{L}}_{\theta_0^*} - \alpha \boldsymbol{\nabla}_{\theta_0^*})}) \leq 0. \tag{9}$$

From (8) and (9), we can easily arrive our conclusions in (6). $\square$

## B  Model Architectures

### B.1  Descriptions of the Meta-model

In miniImagenet, we use the classical convolutional neural network with four convolutional layers. In drug activity prediction, the base model is set as two fully connected layers with an additional linear layer for prediction, where each fully connected layer consists of 500 hidden units and LeakyReLU is used as the activation function.

### B.2  Descriptions of the Neural Scheduler

Figure 1 presents the architecture of the neural scheduler. We apply one fully connected layer with hidden dimension 5 to embed the percentage of training iteration. Bidirectional LSTM with hidden units set to 10 is used for embedding both the loss and the gradient similarity, due to its remarkable expressive power. Afterwards, we concatenate all the embedded features in the fusion layer.

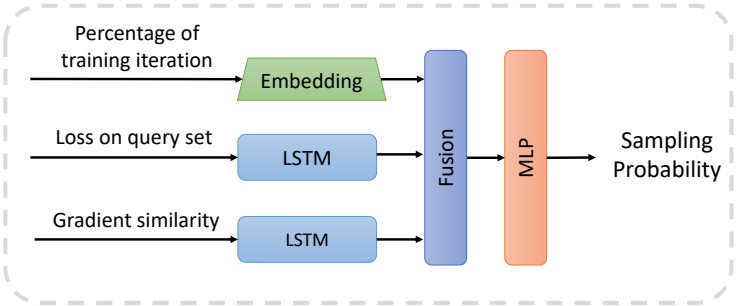

Figure 1: Illustration of the architecture of the neural scheduler $\phi$.

## C  Experimental Setups for Meta-learning with Noise

In miniImagenet, we create the noise tasks by applying the symmetry flipping [1] on the labels of the support set. Figure 2 illustrates the noise transition matrix, where the diagonal elements (i.e., the elements with dark color) denote the probability of keeping the original labels and the off-diagonal elements (i.e., the elements with light color) represent the probabilities of flipping to that class. Note that the noise ratio considered in Figure 2 is 0.8. The procedures of creating noisy tasks for drug activity prediction have been provided in the main paper.

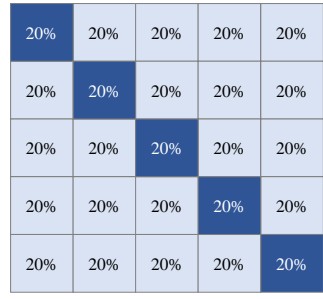

Figure 2: Illustration of flipping probabilities for symmetry flipping.

### C.1  Hyperparameters

**miniImagenet**: In our experiments, we empirically find that it is non-trivial to jointly train the meta-model and the neural scheduler in this noisy task setting. Thus, we first train the neural scheduler $\phi$ for 2,000 iterations without updating the meta-model $\theta_0$. The learning rate of training the neural scheduler $\phi$ is set as 0.001.

After training the neural scheduler for 2,000 iterations, we iteratively optimize the neural scheduler $\phi$ and the meta-model $\theta_0$. The maximum number of meta-training iterations is set as 30,000. For noisy tasks, we also introduce the warm-start strategy as mentioned in [2] – a pre-trained meta-model from clean data is used in the beginning. For other hyperparameters, we set the meta batch size, the query set size as 2 and 15, respectively. During meta-training, the number of inner-loop updates is set as 5. The inner-loop and outer-loop learning rates are set as 0.01 and 0.001, respectively. For ATS, the pool size of our tasks is set as 10 when the noise ratio is in $\{0.2, 0.4, 0.6\}$, and is set as 15 if noise ratio amounts to $0.8$. The temperature of the softmax function for outputting sampling probabilities in the neural scheduler is set to be $0.1$. The gradient similarity input for the neural scheduler includes cosine similarity between the gradient of meta-training loss and meta-testing loss, the gradient norm.

**Drug activity prediction**: For the drug activity prediction, the meta batch size is set as 8 and the size of the candidate task pool is set as 20. We alternatively train $\phi$ and $\theta$ from scratch. The maximum number of meta-training iterations is set as 84,000. The inner-loop and out-loop learning rates are set as 0.01 and 0.001, respectively. Cosine similarity is used to calculate gradient similarity.

## D  Results for Meta-learning with Limited Budgets

### D.1  Experimental Settings

For miniImagenet, we conduct the experiments by controlling the number of meta-training classes. The selected 16 classes under this setting are: {n02823428, n13133613, n04067472, n03476684, n02795169, n04435653, n03998194, n02457408, n03220513, n03207743, n04596742, n03527444, n01532829, n02687172, n03017168, n04251144}.

In addition, we analyze the influence of the budgets by increasing the number of meta-training classes. In this analysis, we list the selected 32 classes to be {n03676483, n13054560, n04596742, n01843383, n02091831, n03924679, n01558993, n01910747, n01704323, n01532829, n03047690, n04604644, n02108089, n02747177, n02111277, n01749939, n03476684, n04389033, n07697537, n02105505, n02113712, n03527444, n03347037, n02165456, n02120079, n04067472, n02687172, n03998194, n03062245, n07747607, n09246464, n03838899}, and the selected 48 classes to be {n02074367, n02747177, n09246464, n02165456, n02108915, n03220513, n04258138, n01770081, n03347037, n01910747, n02111277, n04296562, n01843383, n02966193, n03924679, n04596742, n03998194, n07747607, n07584110, n03207743, n04515003, n03047690, n04435653, n04604644, n02457408, n04251144, n04509417, n02091831, n04243546, n02108551, n03062245, n13133613, n13054560, n06794110, n04612504, n03400231, n02606052, n04275548, n01749939, n02108089, n03888605, n02120079, n04443257, n04389033, n03337140, n02823428, n03476684, n04067472}, respectively. The setting with 64 classes corresponds to the original miniImagenet dataset.

## D.2 Hyperparameters

**miniImagenet**: In this experiment, the learning rate of training the neural scheduler $\phi$ is set as 0.001. We alternatively train the meta-model $\theta_0$ and the neural scheduler $\phi$ from the beginning and set the maximum number of meta-training iterations as 30,000. The meta batch size is set as 2, and the size of the candidate task pool is set as 6. We set the temperature of the softmax function for producing sampling probabilities in the neural scheduler as 1. The input for the scheduler is the same as described in the noisy task setting.

**Drug activity prediction**: In drug activity prediction, the hyperparameter settings under the limited budgets scenario are the same as the noisy task scenario.