# OpenReview forum: "Meta-learning with an Adaptive Task Scheduler"
_NeurIPS.cc/2021/Conference — NeurIPS 2021 Poster_

### Official Review · Reviewer_nPYJ · 2021-07-16

**Rating:** 7
**Confidence:** 4

**Summary:**

This paper proposed Adaptive Task Scheduler (ATS) for meta-learning, which trains a neural network to adaptively select tasks. A bi-level optimization strategy is used to optimize the the meta-model and neural scheduler. Experiments are conducted on miniImagenet  and drug compounds, which beats the baseline methods.

**Limitations And Societal Impact:**

I think the story brought by Proposition 1 is questionable. Indeed, certain sampling probability $w$ could make the meta-training loss lower. However, meta-training loss is not our ultimate goal. Our goal is the performance on the test set. In some extreme cases where some tasks available in the training phase have very low noise, such that one would give them high sampling probability to reduce the meta-training loss. However, they might be unrelated to some tasks in the test phase. So I am not sure how meaningful is Proposition 1.

**Main Review:**

The paper is a good attempt on the adaptive task scheduler. The proposed algorithm totally make sense to me. In general, the paper is well-written and easy to follow. The experimental results are nice.


Typos:
Line 140: computationally
Equation (6): the definition of $\theta$ should be $\theta_{i}^{(k)}$
Equation (7): should be $\theta_{i}^{(k)}$ in the bottom equation.
Equation (9): should be $\theta_{i}=\theta_{0}-\alpha \nabla_{\theta} \mathcal{L}\left(\mathcal{D}_{i}^{s} ; \theta_0\right)$

**Time Spent Reviewing:**

4

---

> ### Author Response · Authors · 2021-08-10
> **Response to Reviewer nPYJ**
>
> Thanks a lot for the constructive comments. We have fixed the typos in our paper, and we explain the significance of Proposition 1 below.
>
> #### **Q1**: The significance of Proposition 1
>
> > According to the generalization bound of meta-algorithms with S/Q episodic training strategy in Theorem 4 of [1], the expected risk of meta-testing tasks is upper-bounded by 1) the average empirical risk on meta-training tasks (i.e., the **meta-training loss**) and also 2) a term at the order of $O(1/\sqrt{n})$ where $n$ is the number of meta-training tasks. Therefore, improving the meta-training loss would definitely improve the expected risk of meta-testing tasks given a fixed number of meta-training tasks, as long as the assumption for Theorem 4 in [1] holds, that is, meta-training tasks and meta-testing tasks are from the same task distribution.
> >
> > [1] Chen, Jiaxin, et al. "A closer look at the training strategy for modern meta-learning." NeurIPS 2020.

---

> > ### Comment · Reviewer_nPYJ · 2021-08-15
> > **Task distributions**
> >
> > I don't agree with the argument. In algorithm 1, the meta-training tasks are sampled either via $W^{(k)}$ or from the updated task scheduler. Both of them are updating, there is no guarantee that the meta-training tasks and meta-testing tasks are from the same task distribution. How do you avoid the algorithm simply selecting simple tasks by putting more weights on them and the hard tasks may still appear in the test phase?

---

> > > ### Author Response · Authors · 2021-08-18
> > > **Response to Reviewer nPYJ about Task Distributions**
> > >
> > >
> > > Thank you for letting us know your concern. We would like to clarify the independence between the change in difficulty level and the shift in task distribution.
> > >
> > > > - The change of difficulty level does not necessarily mean that the distribution shifts. We draw an analogy with boosting and curriculum learning algorithms where **samples** are weighted according to the difficulty level. However, as evidenced in the generalization bound in [1] and the experimental setup in [2], the distribution remains the same across the training set and the testing set. For example, hard examples could be those around the decision boundary but within the same distribution.
> > > >
> > > > - Following the standard protocol of meta-learning [3] and analogous to sample curriculum learning, our model is built upon the general task distribution setting where both meta-training and meta-testing tasks are i.i.d. sampled from **the same task distribution** (e.g., miniImageNet). Therefore, Theorem 4 mentioned in the response to your Q1 holds -- improving the meta-training loss will improve the expected risk of meta-testing tasks.
> > > >
> > > > -  Under the same task distribution, we empirically validate the effectiveness of ATS by varying the difficulty level of meta-testing tasks.
> > > >
> > > >     - For clean meta-testing tasks, extensive experiments have validated the effectiveness of ATS in our paper.
> > > >
> > > >     - For noisy meta-testing tasks, the results under miniImagenet-noisy 1-shot setting are reported in the below table, verifying that ATS can still improve the performance even if there are "hard tasks" among meta-testing tasks.
> > > > | Ratio of Noisy Meta-testing Tasks | Uniform           | BNS (best baseline) | ATS (ours)        |
> > > > |-----------------------------------|-------------------|---------------------|-------------------|
> > > > | 0%                                | 41.67 $\pm$ 0.80% |  42.13 $\pm$ 0.79%  | **44.21 $\pm$ 0.76%** |
> > > > | 10%                               | 39.10 $\pm$ 0.89% |  40.07 $\pm$ 0.93%  | **41.06 $\pm$ 0.95%** |
> > > > | 30%                               | 34.30 $\pm$ 1.10% |  34.89 $\pm$ 1.14%  | **35.96 $\pm$ 1.18%** |
> > > > | 50%                               | 31.05 $\pm$ 1.14% |  31.27 $\pm$ 1.18%  | **32.09 $\pm$ 1.19%** |
> > > >
> > > >[1] Freund, Yoav, Robert Schapire, and Naoki Abe. "A short introduction to boosting." *Journal-Japanese Society For Artificial Intelligence* 14.771-780 (1999): 1612.
> > > >
> > > >[2] Jiang, Lu, et al. "Mentornet: Learning data-driven curriculum for very deep neural networks on corrupted labels." *International Conference on Machine Learning*. PMLR, 2018.
> > > >
> > > >[3] Finn, Chelsea, Pieter Abbeel, and Sergey Levine. "Model-agnostic meta-learning for fast adaptation of deep networks." In International Conference on Machine Learning, pp. 1126-1135. PMLR, 2017.

---

> > > > ### Comment · Reviewer_nPYJ · 2021-08-18
> > > > **Response**
> > > >
> > > > Thanks authors for the detailed discussion. I understand that we might be able to show a generalization bound with the weighted loss as the case in boosting and the experimental performance can also be good. However, my argument is simple: directly minimizing the sample weighted loss does not necessarily imply the expected loss of the original distribution. The story in Proposition 1 is not complete. For example, one can simply divide the response variable of a task by 1000. Then a simple $w$ that put all the weights on this task will significantly reduce the meta-training loss. In other words, having a low meta-training loss can be easily via some trivial method, yet it may lead to a bad generalization performance.

---

> > > > > ### Author Response · Authors · 2021-08-19
> > > > > **Response to Reviewer nPYJ about Proposition 1**
> > > > >
> > > > > Thank you again for re-formulating your concern which we believe boils down to whether the global minimum to the weighted meta-training loss $\mathcal{L}^w(\theta_0)$ and that to the original meta-training loss $\mathcal{L}(\theta_0)$ are the same. Notice that, it seems the Openreview system can not display long latex equations. Please kindly paste the following latex formulations to any latex editors.
> > > > >
> > > > > > - Proposition 1, as stated in Line 167-168, only shows the connection between the weighted meta-training loss $\mathcal{L}^w(\theta_0)$ and the original meta-training loss $\mathcal{L}(\theta_0)$; it does not reach any conclusion on the relationship between the global minimum to $\mathcal{L}^w(\theta_0)$ and that to $\mathcal{L}(\theta_0)$.
> > > > > >
> > > > > > - In fact, Proposition 1 is prepared to Proposition 2 where we reach the following conclusion. **Under the assumption (Line 176-179) that the task scheduler does not change the global minimum**, i.e., $\theta_0^*=\arg\min\mathcal{L}(\theta_0)=\arg\min[Cov(\mathcal{\boldsymbol{L}}_{\theta_0},\mathbf{w})-\alpha Cov(\boldsymbol{\nabla}_{\theta_0},\mathbf{w})]=\arg\min\mathcal{L}^w(\theta_0)$,
> > > > > >
> > > > > >   - the meta-training loss improves as long as the sampling probability $\mathbf{w}$ is negatively correlated with the loss but positively correlated with the gradient similarity between the support and the query set. That is, the global minimum $\theta^*_0$ is more pronounced in this case, i.e.,
> > > > > >
> > > > > >     $\mathcal{L}^w(\theta_0)-\mathcal{L}^w(\theta_0^*)=\mathcal{L}(\theta_0)+Cov(\mathcal{\boldsymbol{L}}_{\theta_0},\mathbf{w})-\alpha Cov(\boldsymbol{\nabla}_{\theta_0},\mathbf{w})-\mathcal{L}^w(\theta_0^*)\geq \mathcal{L}(\theta_0)+Cov(\mathcal{\boldsymbol{L}}_{\theta_0^*},\mathbf{w})-\alpha Cov(\boldsymbol{\nabla}_{\theta_0^*},\mathbf{w})-\mathcal{L}^w(\theta_0^*)=\mathcal{L}(\theta_0)-\mathcal{L}(\theta_0^*)$.
> > > > > >
> > > > > >   - the optimization landscape is improved (Line 182-187) in that 1) for those parameters $\theta_0$ that are far from the optimal meta-model $\theta^*_0$, the gradients become overall steeper for speed-up, and 2) for those parameters $\theta_0$ that are within the variance of the optimal meta-model $\theta^*_0$, the minima tends to be flat with **better generalization ability**.
> > > > >
> > > > > We will definitely follow the reviewer's suggestion to re-organize the assumptions and propositions to avoid any misunderstanding here.

---

> > > > > > ### Comment · Reviewer_nPYJ · 2021-08-19
> > > > > > **Response**
> > > > > >
> > > > > > Thanks the authors for providing the additional clarifications. I am sorry I missed the assumptions ($\theta_0^*=\arg\min\mathcal{L}(\theta_0)=\arg\min[Cov(\mathcal{\boldsymbol{L}}{\theta_0},\mathbf{w})-\alpha Cov(\boldsymbol{\nabla}{\theta_0},\mathbf{w})]=\arg\min\mathcal{L}^w(\theta_0)$).
> > > > > >
> > > > > > I sincerely suggest the authors to give this assumption a separate block as to my understanding this is a very strong assumption and there should be more discussions on this assumption. To my understanding, this assumption directly assumes that a weighed loss does not change the global minima of the original loss. To shel some lights on when the assumption can hold, let us consider simple linear regressions. It requires $\boldsymbol{\nabla}_{\theta_0}[Cov(\mathcal{\boldsymbol{L}}{\theta_0},\mathbf{w})-\alpha Cov(\boldsymbol{\nabla}_{\theta_0},\mathbf{w})] \mid_{\theta_0 = \theta_0^*} = 0$. If we do not do the extra updates for each task and assume the covariate distribution are the same across the domain, the second term will be almost the same across the tasks (equal to the covariance matrix of the covariates).Then $w$ has to be uniform to keep the same optima. It seems to me that $w$ adjust to the covariate shift in this case. In general, if one can show that the $w$ that makes the assumption hold coincidences with the $w^*$ proposed in Proposition 2, it makes an interesting result
> > > > > >
> > > > > > I appreciate the authors for the interesting discussion. I decide to keep my original score.

---

### Official Review · Reviewer_jUKV · 2021-07-16

**Rating:** 5
**Confidence:** 4

**Summary:**

The author proposed a dynamic sampling approach to form meta learning tasks, with the sampling distribution determined by 1) loss of the tasks, 2) the similarity between gradient of support and query data. Experimental results on the setting of noisy meta learning and limited budgets demonstrate the effectiveness of the proposed approach.

**Limitations And Societal Impact:**

The authors discussed limitations.

**Main Review:**

1)   I agree on the author’s intuition which is to sample out tasks that  are informative and not outliers. The author used loss to determine on the difficulty and used the alignment of gradient between support and query data to determine outlier. However, this alignment of gradient can’t infer the task is not an outlier, i.e., when the tasks are from more than one domain, task of another domain can align perfectly by itself but still of little value to the current domain. The algorithm design needs improvement to cover such cases.

2)   The author used LSTM stacks to process on the two factors. Given meta learning models are usually light weight by itself, it would be beneficial to know on the computational cost from this added component and how it impacts the training efficiency.


3)    Better to add a little description on motivation of adding label noise in the data. i.e. will this make it more close to real world scenarios? Or it is for a fair comparison with previous works and less bias prone? etc.

4)  The paper works on noisy meta learning setting and demonstrate the effectiveness of the proposed approach. I would like to know what is the performance on the normal meta learning setting without noise.




**Time Spent Reviewing:**

5 hours

---

> ### Author Response · Authors · 2021-08-10
> **Response to Reviewer jUKV**
>
> We greatly appreciate your comments on this paper. Below, we address your concerns point by point. Please kindly let us know whether you have any further concerns.
>
> #### **Q1**: the alignment of gradient cannot infer the task is not an outlier
>
> > - As stated in Line 47-50, the neural scheduler learns the sampling probability of a task by **simultaneously** considering 1) the loss of the meta-model on this task and 2) the gradient similarity between the support and query sets of this task, which characterize the difficulty of this task from the perspectives of learning outcome and learning process, respectively.
> >
> > - The neural scheduler **taking both factors into consideration** is capable of handling the following four cases, where it assigns
> >
> >   - large weights for those tasks belonging to case (1) which are desired;
> >   - small weights for those tasks belonging to case (2) which the reviewer refers to, i.e., out-of-domain tasks. Though they can align perfectly with a large gradient similarity, they have a large loss as they are too distant for the meta-model to adapt to.
> >   - small weights for those tasks belonging to case (3), which are memorized tasks, where the meta-model directly memorizes the query set without adaptation from the support.
> >   - small weights for those tasks belonging to case (4), which are true hard tasks, where it is hard to generalize from the support to the query set due to distributional shift.
> >
> >   |                               | small loss | large loss |
> >   | ----------------------------- | ---------- | ---------- |
> >   | **large gradient similarity** | (1)        | (2)        |
> >   | **small gradient similarity** | (3)        | (4)        |
>
> #### **Q2**: Training efficiency
>
> > Compared with uniform sampling, the running time of ATS is around three times slower. The training process of ATS can be finished within 12 hours for both miniImagenet and drug datasets. We would like to emphasize that our main focus in this work is to improve the performance of meta-learning under the setting of meta-learning with noise and limited budget. We will continue to investigate how to tackle this efficiency problem in the future.
>
> #### **Q3**: Motivation of adding label noise in the data
>
> > - In Line 25-27, we have illustrated one of the motivations for equipping the meta-learning framework with a task scheduler, i.e., there are noisy tasks in real-world scenarios.
> > - In Line 53-61, we have also highlighted one of our contributions in that the proposed task scheduler indeed corroborates our motivation in dealing with noisy tasks.
> > - The label noise is introduced to reflect different magnitudes of label corruption in real-world applications, which are widely adopted in the previous literature [1].
> >
> > [1] Charoenphakdee, Nontawat, Jongyeong Lee, and Masashi Sugiyama. "On symmetric losses for learning from corrupted labels." ICML 2019.
>
> #### **Q4**: normal meta-learning setting without noise
>
> > We have conducted the experiments on meta-learning without noise, which could be regarded as meta-learning with full budget (see Table 5 with class 64 (full budget)). We also conduct additional experiments to show the effectiveness of ATS in the normal miniImagenet setting by incorporating ATS with different backbone meta-learning methods. Please also kindly refer to the response of Q2 and Q3 for Reviewer VBvz.

---

> > ### Comment · Reviewer_jUKV · 2021-08-18
> > **response**
> >
> > Thank you for your reply
> >
> > (1)  small weights for those tasks belonging to case (2) which the reviewer refers to, i.e., out-of-domain tasks. I think this argument is not convincing. Out of domain task are not always large loss, this depends on how hard are the out of domain tasks. For example, the in-domain is miniimagenet, and out of domain is aircraft, omniglot. The proposed rule only covers a small portion of the real scenarios.
> >
> > (2)   Traning efficiency is kind of slow, this is a concern for deployment.
> >
> > (3)   In normal meta learning, the improvement seems marginal. This raises the question that in real scenairos, the problem could be mixed of meta learning without noise and some cases with noise. In this case, the performance may not improve a lot.
> >
> > (4)   I recently find similar spirit for task scheduling, https://arxiv.org/pdf/1702.08635.pdf, this proposes similar method to this paper for standard deep learning.
> >
> > In summary, I have to keep my score.

---

> > > ### Author Response · Authors · 2021-08-19
> > > **Response to Reviewer jUKV’s Additional Concerns**
> > >
> > > Thank you for letting us know your further concerns. Below, we reply to your concerns point by point.
> > >
> > > - (1) Out-of-domain tasks usually lead to larger losses. Here, under the original 1-shot setting, we use the model pre-trained under miniImagenet and compute the loss on both miniImagenet and aircraft. Among 600 tasks, the loss values of 595 tasks are larger than 1.3, while the number reduces to 67 in miniImagenet. Thus, the results verify our assumption.
> > >
> > > - (2) As discussed in our initial response to Q2, our main focus in this paper is to improve the performance and we will investigate how to improve the meta-training efficiency in the future. Besides, the meta-testing cost is the same as original meta-learning, which could be regarded as a more important factor in deployment.
> > >
> > > - (3) In our experiments, we have conducted the experiments under the mix setting (please kindly refer to the discussion in Line 226 of the main paper), and reported the results in Table 2 and Table 3. In particular, the results in Table 3 verify that ATS can improve the performance under different noise ratios.
> > >
> > > - (4) We would like to point out that i). the problem is entirely different between our paper and [1]; ii). designing the task scheduler in meta-learning is more challenging than the sampling strategy in supervised learning. In meta-learning, we need to consider the correlations during the bi-level optimization process and characterize the difficulty of a task.
> > >
> > >
> > > [1] Fan, Yang, et al. "Learning what data to learn." arXiv preprint arXiv:1702.08635 (2017).

---

### Official Review · Reviewer_VBvz · 2021-07-16

**Rating:** 6
**Confidence:** 5

**Summary:**

Meta-learning is a popular strategy to learn a well-generalized model from different meta-training tasks. Classical meta-learning assumes equal importance for all the meta-training tasks. Recent work has proposed sampling of the tasks based on various factors such as difficulty, amount of information, etc. The paper presents an extension to these ideas by proposing an adaptive task scheduler. In contrast to the prior approaches where the task distribution is hard-crafted and fixed, the authors propose a learnable task scheduler that outputs the sampling probability score of each task in a task pool to directly minimize the generalization error. The inputs to the task scheduler - the query loss and the similarity between the gradients on the query and support set are theoretically grounded and well-motivated. The meta-model and the tasks scheduler are learned using a bi-level optimization; in particular, REINFORCE is used to update the task scheduler. The authors conduct experiments on the mini-imagenet (classification) and drug discovery (regression) datasets to validate the effectiveness of the task scheduler. The results indicate the robustness of the task scheduler on both the datasets under noisy conditions and limited task budget.


**Limitations And Societal Impact:**

yes

**Main Review:**

Meta-learning is among the popular approaches for learning generalizable models. An aspect of this learning paradigm that could be further optimized was task selection. While the recent approaches along this direction use a fixed or hand-crafted method to select the tasks, this paper proposes a novel method to learn the distribution automatically. This is a valuable contribution. I liked the theory motivating the two factors input to the automatic task scheduler. This is the strongest contribution of the paper. I am sure that this will spawn new focussed research on improving the task scheduler for meta-learning. Having said that, there are a couple of issues with the methodology as discussed below.

Randomly selecting $N^{pool}$ of tasks is understandable due to computational constraints. However, I do not see the need for a second round of task selection from this candidate pool based on the estimated weights. Why cannot the meta-model be trained using all the $N^{pool}$ tasks? According to Section C.1, the meta-batch size is 2 and $N^{pool}=10$. Is there any issue with $N^{pool}=4$ and the meta-batch containing all the tasks in the $N^{pool}$? I ask this as the theory seems to be more aligned with weighted task loss rather than sampled task loss.

Experiments on the noisy mini-imagenet and drug discovery datasets validate the effectiveness of the ATS. The ablation studies are also adequate. However, there are two issues with the experimental setup, as elaborated below.

- The authors restrict the evaluation to only the noisy version of the mini-imagenet. While ATS might work best on noisy data, an optimal ATS module should work on any generic meta-learning task (even ones without noise). The bare minimum would be to validate ATS on the original mini-imagenet, tiered-imagenet, and CIFAR-FS benchmarks widely adopted in the literature. Table 3 suggests that ATS does not yield significant improvements with reduced noise levels. Thus, it is natural to gauge the performance on a ‘noiseless’ dataset. No noise in the dataset does not mean the tasks are easily learnable. The suggested benchmark datasets are extremely challenging and are perhaps hard learning tasks. This will test the performance of ATS on truly hard tasks. I agree that ATS might perform better on highly noisy datasets (noise level of 60% and above), but such settings as illustrated in the paper are quite unrealistic.

- ATS is compared only against other sampling strategies. In my opinion, this is an unfair comparison. Liu et al. eccv 2020 [16] - GCP; compares the performance of various meta-learning approaches on the standard benchmark datasets using different backbones. I believe it is necessary to compare the performance of the ATS with some of these approaches. This will address a fundamental question - is task selection critical for meta-learning? The paper implicitly assumes it and does not present compelling evidence. It would be worthwhile to adopt and compare the sampling strategies on meta-learning approaches that allow for it, instead of restricting only to the ANIL framework.

In my humble opinion, addressing the following comments will further strengthen the contribution.

1. The results on all the datasets require additional significance tests. This is especially true with the drugs dataset where the difference in the mean $\mathcal{R}^2$ is in the second decimal place.
2. Line 268 - Are the weights for the 1000 meta-training tasks estimated at the same time by the ATS? If so, at what stage in the training process is these weights estimated?
3. The authors mention in the supplementary material that warm-start initialization refers to the meta-model that is pre-trained using the original datasets (without noise). I think this is unfair, as the meta-model has already seen the clean meta-training tasks. The proposed training process is only trying to maintain the performance. This also negates the primary assumption of the paper - training with noisy and hard tasks. I believe a fair comparison warrants training from scratch on the noisy datasets directly.
4. Line 231 mentions a set of clean validation tasks for ATS. What is the role of this validation set? This can be further added to Algorithm 1.
5. Would you please explain the pre-training process of a neural scheduler? Is the meta-model $\theta_0$ used during pre-training randomly initialized or pre-trained? If $\theta_0$ is pre-trained, is it pre-trained along the lines of supervised learning on meta-train set or episodic regimen? Does pre-training of neural scheduler $\phi$ follow episodic regimen? Does pre-training of $\phi$ and $\theta_0$ both occur on clean data? Is pre-trained $\theta_0$ used as such (warm start), or $\theta_0$ is reinitialized during the iterative optimization phase?

The paper is well structured and written. The ideas are easy to understand and well-articulated.

Corrections to the theory in the supplementary material----

- In the proof of proposition 1, we are adding and subtracting two terms $\mathcal{L(D_i^q;\theta_i})$ and $\sum_i^{N^{pool}}w_i\mathcal{L}(D_i^q;\theta_i)$. The second term in the first step of the proof should be $-\sum_{i=1}^{N^{pool}}w_i(\mathcal{L}(D_i^q;\theta_i)-\mathcal{L}(D_i^q;\theta_i))$

- Also, the second order term in equation 3 of the supplementary material is incorrectly stated (Though it does not impact the rest of the analysis). It should be $\alpha^2 \nabla_{\theta_0}\mathcal{L}(D_i^s;\theta_0)^T\nabla_{\theta_0}^2\mathcal{L}(D_i^q;\theta_0)\nabla_{\theta_0}\mathcal{L}(D_i^s;\theta_0)$

Typos

Line 141 - differentiate - differentiable

Line 203 - follow - following

Post response************
I thank the authors for providing a detailed response to my queries.

I am satisfied with the authors' response to most of the questions. I still have a few lingering questions/suggestions that will further add clarity to the results

Q2 and Q3 I am not convinced that the results in Table 5 represent the full budget setting. The accuracies for MAML in the 5.1 and 5.5 settings stand at 48.7% and 63.1% respectively as reported in the literature. However, the table reports only 45.7% accuracy. Furthermore, in the response to Q3, the authors state MAML performance as 47.09% and 63.1% for the two settings. So, why are the accuracies for the classical MAML different at different places within the paper?

Q4 - I am assuming that the term after $\pm$ is the standard deviation. Significance tests do not mean only reporting the standard deviation, but checking the p-value for this the results are significant. At the moment, no such test has been performed.

Overall, I am inclined to raise my rating of the paper.

**Time Spent Reviewing:**

8

---

> ### Author Response · Authors · 2021-08-10
> **Response to Reviewer VBvz**
>
> We sincerely appreciate your constructive comments to improve our paper. We detail our response below and have corrected the typos in our paper. Please kindly let us know if our response addresses the issues you raised in the paper.
>
> #### **Q1**:  weighting with $N^{pool}=4$ and meta_batch_size $=4$ vs. sampling with $N^{pool}=10$ and meta_batch_size $=2$
>
> > - We have indeed tried weighting all tasks in a meta batch ([setup: $N^{pool}=4$ and meta_batch_size $=4$]) before, while the results are not as competitive as sampling from a large pool of candidate tasks. The results are:
> >
> > |             | miniimagenet-noisy |                   | Drug-noisy |        |      |
> > |-------------|:------------------:|:-----------------:|:----------:|--------|:----:|
> > |             | 5-way 1-shot       | 5-way 5-shot      | mean       | medium | >0.3 |
> > | Weighting   |  43.60 $\pm$ 0.79% | 56.82 $\pm$ 0.68% |    0.224   | 0.133  |  28  |
> > | **ATS**         |  **44.21 $\pm$ 0.76%** | **57.68 $\pm$ 0.70%** |    **0.233**   | **0.152**  |  **31**  |
> >
> > - We attribute such performance gap between the two methods to **the number of effective tasks**.
> >   - Weighting: meta_batch_size $\leq 4$ is widely adopted in meta-learning due to memory constraints, so that setting $N^{pool}=$ meta_batch_size leads to a less diverse candidate pool. If the randomly sampled tasks are all noisy, the number of effective tasks would likely be less than the meta_batch_size.
> >   - Sampling: By setting a larger size of $N^{pool}$ (e.g., 10) and sampling the top meta_batch_size tasks according to the predicted probability,  the number of effective tasks would be likely equal to the meta_batch_size.
>
> ####  **Q2**: Performance on a generic meta-learning task
>
> > We have conducted the experiments on the generic meta-learning task, which could be regarded as meta-learning with a full budget. We have reported the results of the generic meta-learning task in Table 5 with class 64 (full budget).
>
>
> ####  **Q3**: Comparison of the ATS with other meta-learning approaches
>
> > - We have indeed validated the assumption that task selection is critical for meta-learning. In our experiments, "Uniform" in Table 1,3,4,5 denotes the vanilla ANIL approach where tasks are randomly sampled with a uniform probability.
> >
> > - Following the reviewer's suggestion, we have also equipped the other two meta-learning approaches with the proposed sampling strategy besides ANIL and compared the performance of different sampling strategies on miniImageNet. Notice that we do not find the authors of GCP paper release their code, and thus we implement GCP by ourselves. All models are evaluated in the same environment. Results show that the proposed ATS is compatible with more meta-learning approaches than ANIL and demonstrates its superiority over GCP.
> >
> >   | Model            | miniImageNet 5-way 1-shot | miniImageNet 5-way 5-shot |
> >   | ---------------- | ------------------------- | ------------------------- |
> >   | MAML             | 47.09 $\pm$ 0.76%           |      63.15 $\pm$ 0.67%       |
> >   | MAML with GCP    | 46.92 $\pm$ 0.83%           |        63.28 $\pm$ 0.66%      |
> >   | MAML with ATS    | 47.89 $\pm$ 0.77%           |      64.07 $\pm$ 0.70%                     |
> >   | MetaSGD          |       48.15 $\pm$ 0.76%                    |        64.03 $\pm$  0.68%                  |
> >   | MetaSGD with GCP |         47.77 $\pm$ 0.75%                 |            63.50 $\pm$ 0.71%               |
> >   | MetaSGD with ATS |          **48.59 $\pm$ 0.79%**                 |            **64.79 $\pm$ 0.74%**               |
>
> ####  **Q4**: Significance tests
>
> > - For all experiments regarding the image datasets, we have provided the significance tests evidenced in Table 1,2,3,4.
> >
> > - For the experiments regarding the drug datasets, we have followed the reviewer's suggestion by adding significance tests in Table 1. We will update all tables with significance tests in the final version.
> >
> >   | Model   | Drug-noisy mean | Drug-noisy medium | Drug-noisy > 0.3 |
> >   | ------- | --------------- | ----------------- | ---------------- |
> >   | Uniform |         0.202 $\pm$ 0.003        |           0.113 $\pm$ 0.002        |           21 $\pm$ 0.431       |
> >   | PAML    |          0.204 $\pm$ 0.003     |          0.120 $\pm$ 0.003        |         24 $\pm$ 0.493        |
> >   | ATS     | 0.233 $\pm$ 0.003      |  0.152 $\pm$ 0.004       | 31 $\pm$ 0.632          |
>
>
> #### **Q5**: Line 268: weights for the 1000 meta-training tasks
>
> > - Yes, the weights for the 1000 meta-training tasks are estimated at the same time by the ATS.
> > - In terms of the stage, we use the iteration after the convergence of the neural scheduler and the meta-model.
>
> ####  **Q6**: Warm-start
>
> > - For miniImagenet, pre-training of the meta-model serves only for "warm-start" so that the meta-model together with the neural scheduler converges much faster, while it does not significantly affect the final performance of ATS. We list the results of ATS without warm-start.
> >
> >   | Model              | miniImageNet-noisy 5-way 1-shot | miniImageNet-noisy 5-way 5-shot |
> >   | ------------------ | ------------------------------- | ------------------------------- |
> >   | ATS w/o warm-start |    43.75 $\pm$ 0.79%                             |       58.11 $\pm$ 0.72%                          |
> >   | ATS                | 44.21 $\pm$ 0.76%                 | 57.68 $\pm$ 0.70%                   |
> > - For drug dataset, we do not perform the pre-training process on it.
>
> ####  **Q7**: A set of clean validation tasks for ATS in Line 231
>
> > - In Line 133-139, Figure 1 and Algorithm 1 (Line 10), we have detailed that ATS aims to learn an optimal neural scheduler by minimizing the average loss of the learned meta-model on $N_v=2$ validation tasks.
> > - $N_v$ validation tasks are sampled from either the noisy tasks or a small set of clean tasks to improve the robustness of the learned neural scheduler. Particularly, for fair comparison, we have also fine-tuned all other baselines using other sampling strategies on the small set of clean tasks as stated in Line 231-232.
>
> ####  **Q8**: Pre-training of the meta-model and the neural scheduler
>
> > In the miniImagenet-noisy, we pre-trained the meta-model with clean data as the "warm-start". Then, the neural scheduler is trained with the pre-trained meta-model $\theta_0$ on the noisy data for the predefined warm-up steps.
> > - Note that in the setting of meta-learning with limited budget, the meta-model $\theta_0$ is randomly initialized and no other data is used.
> > - During each iteration of the warm-up training process of neural scheduler $\phi$, we virtually update the meta-model $\theta_0$ with the training tasks, and then evaluate its performance on the validation tasks. With these performances, we can improve our scheduler $\phi$ via Policy Gradient. After this iteration, we reset the model $\theta_0$ to its original parameters (before updating the neural scheduler).
> >
> >In summary, we perform warmup training of the neural scheduler on the noisy (limited) datasets, and the meta-model $\theta_0$ is not updated during this process. After the pre-training of the neural scheduler, we use the pre-trained $\phi$ and the model $\theta_0$ together.

---

> > ### Author Response · Authors · 2021-08-18
> > **Response to Reviewer VBvz's Additional Questions about Q2, Q3, Q4**
> >
> > Thank you a lot for letting us know your concerns so quickly. Here we provide a follow-up response to the remaining two concerns.
> >
> > #### **Q2 & Q3**: Performance gap between original MAML paper and the reported results
> >
> > > - We would like to point out that the reported results in Table 5 with full budget is evaluated on ANIL under the generic meta-learning tasks. The results for MAML under the full budget setting are reported in the table in the response to your Q3.
> > > - The performance gap between the original MAML (48.7% and 63.1%) and our reported results in the table in the response to your Q3 (47.09% and 63.15%) is caused by the randomness in selecting the meta-testing tasks. In the original MAML paper, the authors randomly select 600 meta-testing tasks to evaluate the performance. We re-run their code in our server and get the reported results. For a fair comparison, though, all baselines and ATS are evaluated on the same server so that the same set of meta-testing tasks is used.
> >
> > #### **Q4**: Significance tests and p-value
> >
> > > We have run our experiments ten times and conduct the two-sample t-test between PAML (the best baseline) and ATS. The p-value of mean $R^2$, medium $R^2$ and $R^2>0.3$ is 2e-7, 8e-9, and 6e-4, respectively, demonstrating that ATS significantly improves the performance.

---

> > > ### Comment · Reviewer_VBvz · 2021-08-18
> > > **Reply to the authors response**
> > >
> > > Thank you for providing the additional clarifications. I strongly advise including these details in the paper. This will help the reader understand and appreciate the results.

---

> > > > ### Author Response · Authors · 2021-08-18
> > > > **Thank you for the Reviewer VBvz's Suggestion**
> > > >
> > > > Dear Reviewer VBvz,
> > > >
> > > > Thank you for the suggestion. We will definitely include these details in the revised version.

---

### Official Review · Reviewer_n8ad · 2021-07-16

**Rating:** 6
**Confidence:** 4

**Summary:**

In this paper, the authors propose an adaptive task scheduler to pick the set of tasks that a meta-learner should solve at a given iteration which will improve its generalizability/transferability to downstream tasks rather than picking a set of tasks at random. Instead of choosing a metric/heuristic based approach, the authors propose to use a second network that learns-to-learn the right set of tasks to sample and the overall training progresses via an interleaved (bi-level) optimization procedure. Authors provide empirical evidence based on two datasets that their proposed scheduler ATS performs better than similar existing approaches. The paper also contains some theoretical justifications on why ATS performs better than random sampling.

**Ethical Concerns:**

There are no ethical concerns of their work as far as I understand.

**Limitations And Societal Impact:**

There are no potential negative societal impact of their work as far as I understand.

**Main Review:**

In this paper, the authors propose an adaptive task scheduler to pick the set of tasks that a meta-learner should solve at a given iteration (rather than picking a set of tasks at random) with an aim to improve its generalizability/transferability for downstream tasks. The authors propose to use a learning based approach by using a second network to automatically determine the weightage of each task from the task pool and then picking the top K tasks from that pool. For the meta-learning part, the authors use a standard gradient based learner (ANIL) and for the scheduling-learner part, they use an LSTM based network and use REINFORCE to train the scheduler network. The authors perform experiments on one classification and one regression task and show that their proposed task scheduler performs better than existing ones when it comes to the final test performance. Authors also provide some theoretical justifications as to why their method provides better generalization compared to random selection.

Overall, the work is good and the paper was easy to follow - the narrative and the notations were all clear. The idea also has enough novelty, especially using another network to select the "hardest" tasks (based on support-query difference and query loss) and then using REINFORCE to train that in a DARTS-style (NAS) bi-level setup.

I have a few concerns that I would like authors to address:
*  The authors mention using two LSTMs for the task selector network. Why was LSTM chosen? What is the temporal piece of information that you wanted to capture?
*  The authors perform some ablation studies but there is one that I think is missing - what if, instead of using another network to learn the right task to choose, you just heuristically determine the best tasks by performing some combination of the two metrics selected -- the  query loss and the gradient similarity between the support and query? For example, what happens if you just pick the top K tasks by taking a simple average of these two metrics instead of having another network that takes these as inputs and then learns to pick the top K tasks? Will there be a significant performance difference between these two?
*  To me, it looks like there will be non-trivial computational overhead in adopting this approach vs random sampling. I'd like to see some training time comparison between the methods which will be more informative for a user to decide if the extra computation gain is worth it.
* For MiniImageNet, the authors mention that number of tasks is $4368$ when $16$ classes are chosen because $16  \choose 5$ is $4368$. However, while picking $5$-way tasks, you also need to pick $5$-shots i.e. choosing $5$ images per class from $600$ images. Therefore, number of distinct tasks is way more than $4368$. I would like the authors to clarify why they mention $4368$ as the total number of tasks in this case.
* Finally, pardon my naivety but I do not follow the derivation of proposition 1 (given in appendix). How did you go from line 1 to line 2 in the proof (starting at line 6 in the appendix)? Also, in the manuscript, it should be mentioned that the proof is given in the appendix.

**Time Spent Reviewing:**

5-6  hours

---

> ### Author Response · Authors · 2021-08-10
> **Response to Reviewer n8ad**
>
> Thank you for your constructive comments. You may find our corresponding explanations below for your concerns. We would really appreicate it if you could let us know if you have any further concerns.
>
> #### **Q1**: Why was LSTM chosen?
>
> > - The inputs to the neural scheduler, as stated in Line 206-207, including the loss, the gradient similarity, and the percentage of training iterations for each task.
> >
> > - Specifically, we input $n$ losses and $n$ gradient similarity values over the last $n$ epochs for each task so as to capture the prediction variance [1]. The two LSTMs are designed to process the loss sequence of length $n$ and the gradient similarity sequence, respectively.
> >
> >   [1] Chang, Haw-Shiuan, Erik Learned-Miller, and Andrew McCallum. "Active bias: Training more accurate neural networks by emphasizing high variance samples." NeurIPS 2017.
>
> #### **Q2**: Missing ablation study: pick the top-$k$ tasks by taking a simple average of the loss and gradient similarity
>
> >
> > - We follow the reviewer's suggestion and conduct the experiments on drug activity prediction by ranking tasks according to a simple combination of the loss and the gradient similarity, i.e., Sim/Loss. The reason why we choose the ratio over the average is that those tasks with higher gradient similarity but smaller losses should be prioritized according to Figure 3. The results are reported below:
> >
> > | Ablation Model            | Drug-Noisy | | | Drug-Full | | |
> > | -------------------------  |: --- :| : --- :| : --- :| : --- :| : --- :| : --- :|
> > | | mean | medium | >0.3 | mean | medium | >0.3 |
> > | Ranking by Sim/Loss       |  0.181 | 0.109 | 22 | 0.187 | 0.099 | 24|
> > | ATS ($\phi$ + Loss + Sim) |  **0.233** | **0.152** | **31** |  **0.252** | **0.179** | **36** |
> >
> > According to the Table, we observe that simply selecting the top-$k$ tasks according to the ratio Sim/Loss **significantly underperforms**, because
> >   - the contribution of each metric is hard to manually define, while the neural schedular can automatically learn;
> >   - the contribution of each metric evolves as training proceeds, which has been ignored in such a simple combination but modeled in the neural scheduler with the percentage of training iterations as input.
>
>
> #### **Q3**: non-trival computational overhead
>
> > - In our experiments, the running time of ATS is around 3 times slower than random sampling (i.e., uniform sampling). In general, the training process of ATS can be finished within 12 hours. Nevertheless, we would like to emphasize that our focus is to improve the performance and we will continue to investigate how to tackle this efficiency problem in the future.
>
> #### **Q4**: Why is 4368 the total number of tasks?
>
> > We would underline the difference between an episode and a task.
> >
> > - Episode: by definition in [2], an episode is a few-shot task by subsampling classes as well as data points.
> >
> > - Task: following Page 8 in [3], the number of distinct training tasks is combinatorial for N-way classification, e.g., $\binom{64}{5}=7.6\times 10^6$ for mini-ImageNet.
> >
> > - Two episodes that share the same classes are considered to be the same task, analogous to different batches in traditional machine learning.
> >
> >   [2] Snell, Jake, Kevin Swersky, and Richard Zemel. "Prototypical networks for few-shot learning." NIPS. 2017.
> >
> >   [3] Wang, Haoxiang, Han Zhao, and Bo Li. "Bridging Multi-Task Learning and Meta-Learning: Towards Efficient Training and Effective Adaptation." ICML 2021.
>
> #### **Q5**: line 1 to line 2 in the proof
>
> > We appologize for the typo, as the second term of Line 2 in the proof should be $\sum_{i=1}^{N^{pool}}w_i(\mathcal{L}(\mathcal{D}^q_i;\theta_i)-\mathcal{L}(\mathcal{D}^q_i;\theta_i))$. This is only for preparation of Line 3.

---

### Decision · Program_Chairs · 2021-09-27

**Decision:**

Accept (Poster)

**Comment:**

The idea of the tasks scheduler for meta-learning is interesting. The solution is technically sound and intuitive. The theoretical analysis is also interesting, although it requires more discussions on when the assumption can hold. The experiments are quite exhaustive. The discussion on the computational cost is non-trivial, and more detailed analysis is needed